# Species and cell-type properties of classically defined human and rodent neurons and glia

Xiao Xu[1†], Elitsa I Stoyanova[1], Agata E Lemiesz[1], Jie Xing[1], Deborah C Mash[2], Nathaniel Heintz[1]*

[1]Laboratory of Molecular Biology, Howard Hughes Medical Institute, The Rockefeller University, New York, United States; [2]Miller School of Medicine, University of Miami, Miami, United States

**Abstract** Determination of the molecular properties of genetically targeted cell types has led to fundamental insights into mouse brain function and dysfunction. Here, we report an efficient strategy for precise exploration of gene expression and epigenetic events in specific cell types in a range of species, including postmortem human brain. We demonstrate that classically defined, homologous neuronal and glial cell types differ between rodent and human by the expression of hundreds of orthologous, cell specific genes. Confirmation that these genes are differentially active was obtained using epigenetic mapping and immunofluorescence localization. Studies of sixteen human postmortem brains revealed gender specific transcriptional differences, cell-specific molecular responses to aging, and the induction of a shared, robust response to an unknown external event evident in three donor samples. Our data establish a comprehensive approach for analysis of molecular events associated with specific circuits and cell types in a wide variety of human conditions.

DOI: https://doi.org/10.7554/eLife.37551.001

*For correspondence:
heintz@rockefeller.edu

Present address: †Cerevance, Cambridge, United Kingdom

## Introduction

The ability to genetically target (*Gong et al., 2003*) and molecularly profile specific cell types (*Heiman et al., 2008*) has begun to provide insight into essential features of the mammalian brain that were discovered in the founding studies of Ramon y Cajal over a century ago (*Cajal et al., 1899*). It has been established, for example, that each anatomically distinct, classically defined cell type expresses a set of genes that is characteristic (*Dougherty et al., 2010*; *Doyle et al., 2008*), that these genes confer properties that are essential for specialized cellular functions (*Kim et al., 2008*; *Nakajima et al., 2014*), and that the expression of these genes is highly correlated with cell specific epigenetic states that organize nuclear function (*Kriaucionis and Heintz, 2009*; *Mellén et al., 2012*). Application of these powerful new technologies in mouse models has also led to the realization that environmental influences (*Heiman et al., 2008*; *Shrestha et al., 2015*), internal physiological cues (*Schmidt et al., 2012*), and disease causing genetic lesions (*Fyffe et al., 2008*; *Ingram et al., 2016*) alter gene expression in affected cell types. Despite the pace of advances in experimental systems, fundamental issues of human brain complexity remain unsolved. It is not known, for instance, how many distinct cell types exist in the human brain, how these cell types vary between individuals or across species, whether the process of brain aging is equivalent between cell types, and why mutations in broadly expressed genes can have devastating consequences in one or a few select cell types.

To address these questions, two main approaches have been taken that enable molecular characterization of cell types without the need for transgenic animals. The first involves gene expression

profiling at the level of single cells (*Darmanis et al., 2016*; *Macosko et al., 2015*; *Saunders et al., 2018*; *Thomsen et al., 2016*; *Zeisel et al., 2015*) or single nuclei (*Grindberg et al., 2013*; *Habib et al., 2016*; *Krishnaswami et al., 2016*; *Lacar et al., 2016*; *Lake et al., 2016*). These studies have been instrumental in providing an unbiased description of the diversity of cell types and even subpopulations of cell types in the mouse and human brain. However, the profiles generated from these studies are highly variable, both due to the technical challenge of amplifying signal from a small amount of starting material as well as biological mechanisms such as transcriptional bursting (*Haque et al., 2017*). Additionally, because single cell studies usually profile all cells from a tissue without enriching for cell types of interest, the vast majority of sequences generated from these studies will correspond to common cell types, with only limited sequences corresponding to rare cell types.

A second approach for molecular characterization has been developed to capture specific cell types, usually by cellular labeling followed by fluorescence activated cell sorting (FACS). Because neurons have complex cellular architecture, and techniques to achieve the single cell suspension required for FACS stress and damage neurons, an alternative strategy is to analyze isolated nuclei instead of whole cells (*Matevossian and Akbarian, 2008*). These methods have enabled transcriptional and epigenetic profiling of several cell types during development of both the mouse and human brain (*Ernst et al., 2014*; *Lister et al., 2013*). However, only a few cell types have been characterized using this approach, primarily due to the difficulty of identifying antibodies to nuclear localized proteins that can label the nuclei from cell types of interest with sufficient specificity to enable sorting. To overcome this limitation, we have taken advantage of the fact that the endoplasmic reticulum membrane (ER) is contiguous with the nuclear membrane (*Hetzer, 2010*; *Watson, 1955*). We reasoned that we could expand the number of antibodies for purifying cell-type specific nuclei by targeting ER proteins in addition to nuclear regulatory proteins. Here we demonstrate that antibodies against both ER resident proteins and plasma membrane proteins being translated in the ER can be used for labeling and sorting nuclei from postmortem brains, dramatically expanding the pool of antibodies available for cell specific nuclear isolation.

In this manuscript, we establish a workflow for purifying and molecular profiling of nuclei from specific cell types in the mammalian nervous system without the need for transgenic animals. As a proof of principle, we used antibodies against a variety of different proteins to purify nuclei and profile gene expression and chromatin accessibility from the major cell types in the mouse cerebellum. We demonstrate that this approach can be readily extended to cell types from rat and human brain, and that the cell specific nuclear expression profiles are pure, reproducible, and comprehensive. These profiles enabled us to perform comparative studies of gene expression at a deeper level than previously achieved, and revealed a surprising degree of evolutionary divergence of gene expression profiles in rodent and human brain, even in classically defined, highly conserved cerebellar cell types (*D'Angelo, 2013*; *Eccles, 1967*; *Llinas, 1969*). As an extension to the gene expression studies, we also used purified nuclei to generated genome-wide maps of accessible chromatin for two cell types in mouse and human. Integrating cell-type specific gene expression profiles, epigenetic profiles, and analysis of DNA sequence conservation revealed that genes that change in expression between mouse and human have concordant changes in chromatin accessibility, and have less conserved regulatory sequences compared to genes that are stably expressed across species.

The ability to profile gene expression in defined cell types in the human brain was critical for our studies of aging as our data suggest that molecular mechanisms of brain aging unfold differently in each cell type. Finally, this approach has allowed us to discover a robust, cell-type specific molecular phenotype in a subset of samples. Although it is not clear whether this reflects an unknown clinical condition or changes that arise due to agonal factors or tissue processing, its presence in samples from control donors highlights the need to interrogate the extent of gene expression heterogeneity across control donors prior to comparative analysis with identified clinical conditions. We anticipate that the methodology established in this manuscript will enable the usage of postmortem tissue from common brain banks for correlative studies of human genetic and clinical data with cell-type specific expression, and for identifying as yet unrecognized molecular characteristics of human brain function and dysfunction.

## Results

### A non-genetic method to specifically isolate nuclei from selected cell types

To generate reference nuclear profiles for cell types of interest and to identify general characteristics of nuclear transcriptional profiles, we employed transgenic mice expressing an EGFP-L10a ribosomal protein fusion under the control of cell-type specific drivers (*Figure 1—figure supplement 1A*) (*Doyle et al., 2008*; *Heiman et al., 2008*) for purification of nuclei by FACS (*Kriaucionis and Heintz, 2009*; *Mellén et al., 2012*) (*Figure 1—figure supplement 1B*). RNA-seq was used to generate nuclear expression profiles for five different cell types – cerebellar Purkinje cells, granule cells, or Bergmann glia, or corticopontine or corticothalamic pyramidal neurons (*Figure 1—figure supplement 1C*; *Figure 1—figure supplement 1D*). As expected, as a large proportion of nuclear transcripts are unspliced, we observe that the majority of the reads from our nuclear RNA-seq samples map to introns (*Supplementary file 1*; *Figure 2—figure supplement 1C-D*). Thus, all read quantification for nuclear samples in this manuscript were performed using whole gene annotations. Analysis of these data and published data for three cortical cell types (excitatory neurons, PV interneurons, and VIP interneurons, *Mo et al. (2015)* demonstrated that nuclear RNA expression profiles can both distinguish different cell types and reveal relationships between cell types (*Figure 1—figure supplement 1E*). This finding confirms the utility of nuclear RNA profiles for characterization of CNS cell types.

Next, we generated expression profiles from non-transgenic brains, focusing on several classically defined neuronal and glial cell types that exhibit unique morphology and distribution in the cerebellum (*Figure 1A*). Candidate antibodies for each cell type were chosen from our previously published translating RNA (TRAP) data (*Dougherty et al., 2010*; *Doyle et al., 2008*). To isolate three neuronal cell types (granule, Purkinje, basket) and two glial cell types (astrocytes and oligodendrocytes) from the cerebellum of wild-type mice, we purified and then labeled nuclei (*Figure 1—figure supplement 2A*) using antibodies against one transcription factor (OLIG2), one ER resident membrane protein Inositol 1,4,5-Triphosphate Receptor Type 1 (ITPR1), and two plasma membrane proteins, Sortilin Related VPS10 Domain Containing Receptor (SORCS3) and Glutamate Aspartate Transporter (EAAT1 or GLAST). To facilitate separation of neuronal from glial nuclei, we used in combination with each of these cell-type specific antibodies an antibody against the splicing protein RBFOX3/NeuN. We also used NeuN to positively select granule cells while removing nuclei from contaminating cell types using a combination of ITPR1, SORCS3, and OLIG2 (*Figure 1B*). Imaging of nuclei labeled with these antibodies confirmed the expected distribution of the target proteins – while the nuclear factors NeuN and OLIG2 are localized within the nucleus, specifically in euchromatin, the ER resident protein ITPR1 and the plasma membrane proteins SORCS3 and EAAT1 localize to the ER/nuclear membrane and appear as a ring around the nucleus (*Figure 1E–F*). However, both flow cytometry and imaging of the labeled nuclei yielded a surprising result: although NeuN is not known to label Purkinje nuclei in cerebellar tissue, staining of nuclei using NeuN robustly labels Purkinje nuclei (*Figure 1E*). This is not due to spectral overlap from the Purkinje specific marker, as ITPR1 localizes to the ER/nuclear membrane while NeuN localizes to euchromatin, both as expected. Although we do observe lower levels of NeuN in labeled basket, astrocyte, and oligodendrocyte nuclei, this result suggests that exposure of antibody epitopes might be slightly different between isolated nuclei and tissue sections.

### Nuclear RNA profiles generated from sorted nuclei are cell-type specific

We next produced gene expression profiles for each of the five cerebellar cell types by isolating RNA from the sorted nuclei and performing RNA-seq. To evaluate the purity of these profiles, we checked the expression of known markers for each cell type of interest. In all cases, we observed high expression of markers only in the appropriate cell type. For example, the granule cell markers *Rbfox3* (NeuN) and *Fat2* are highly expressed in granule cells and are minimally expressed in the other four cell types (*Figure 1C–D*). Next, we evaluated these samples at a genome-wide level by computing the pairwise Pearson's correlation coefficient (r) between all samples using normalized gene expression (*Figure 1G*). From this analysis, we could make several observations. First, the

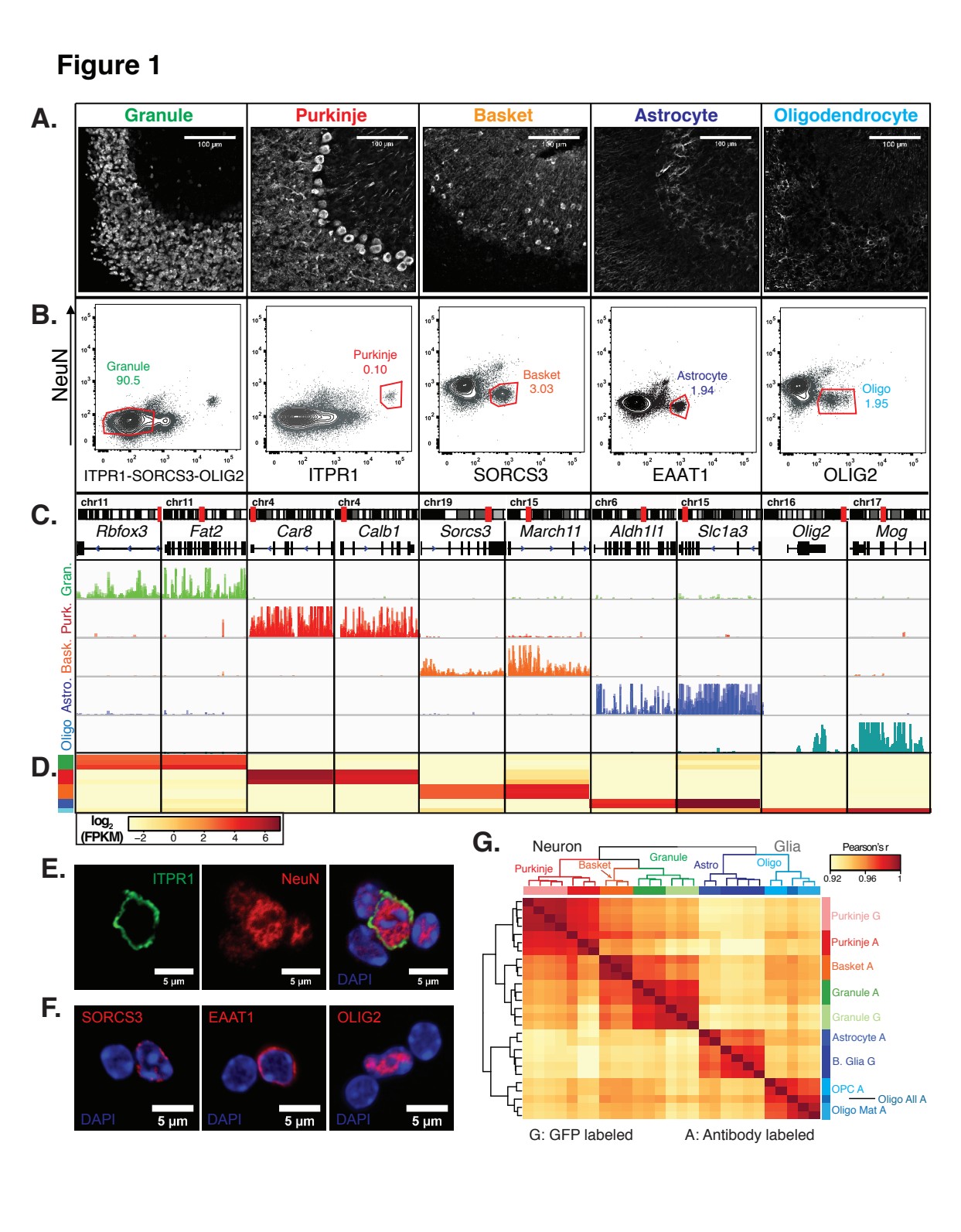

**Figure 1.** Generation of gene expression profiles for distinct cell types from the cerebella of wild-type mice. (**A**) Immunofluorescence staining of five distinct cell types in the cerebellum. Antibodies used to label each cell type: NeuN for granule cells, ITPR1 for Purkinje cells, SORCS3 for basket cells, GFAP labels the cell bodies and process of astrocytes, MOG labels the cell bodies and process of oligodendrocyte. (**B**) Fluorescence activated sorting of stained nuclei from five cell types. Antibodies used for staining are indicated on the x- and y-axes. Percentage of each cell type based on the

*Figure 1 continued on next page*

*Figure 1 continued*

positive population is indicated. (C) Browser view showing examples of gene that are specifically expressed in each of the five cell types. (D) Heatmap of FPKM levels for example genes in each of the five cell types (granule – green, Purkinje – red, basket – orange, astrocyte – blue, oligodendrocytes - cyan). (E-F) Examples of labeled cerebellar nuclei using cell-type specific antibodies. Nuclei are counterstained with DAPI, a marker for heterochromatin. (E) Nuclei labeled with antibodies against ITPR1 and NeuN. Itpr1, an endoplasmic reticulum membrane protein, is localized at the nuclear membrane, while NeuN, a splicing factor, is localized at euchromatin inside the nucleus. (F) Antibodies against the basket cell marker SORCS3 and astrocyte marker EAAT1, two cellular membrane proteins, show labeling of the nuclear membrane. Antibodies against the oligodendrocyte marker and transcription factor OLIG2 show labeling in euchromatin. (G) Heatmap showing the pairwise Pearson's correlation coefficient of GFP- (G) and antibody- (A) sorted nuclei. Hierarchical clustering is performed on the 250 most variable genes across all conditions. See also *Figure 1—figure supplement 1*, *Figure 1—figure supplement 2*.

DOI: https://doi.org/10.7554/eLife.37551.003

The following figure supplements are available for figure 1:

**Figure supplement 1.** Nuclear RNA profiles can specify cell-type identity, related to *Figure 1*.

DOI: https://doi.org/10.7554/eLife.37551.002

**Figure supplement 2.** Overview of nuclei labeling and sorting strategy and comparison to single nuclei sequencing, related to *Figure 1*.

DOI: https://doi.org/10.7554/eLife.37551.004

correlation between different cell types in the cerebellum is high: r = 0.91–0.94. Second, the correlation across biological replicates is very high: r = 0.98–1. Third, when we compared to the reference gene expression profiles from the EGFP-L10a labeled nuclei from granule cells, Purkinje cells, and Bergmann glia, we observed very high correlations between the genetically defined cell types and antibody defined cell types: r = 0.98 for granule cells and r = 0.99 for Purkinje cells; r = 0.96–0.97 between Bergmann glia purified from *Sept4* EGFP-L10a animals and astrocytes purified via antibody labeling (*Figure 1G*). Since Bergmann glia are a sub-population of cerebellar astrocytes, it makes sense for these datasets to have slightly lower correlations. Fourth, when we perform hierarchical clustering, the samples cluster according to the cell class. Thus, neuronal and glial samples separate from each other in the first split. The five cell types then separate from each other regardless of whether the nuclei were purified using genetic labeling (G) or antibody labeling (A). Finally, although the antibody labeled samples undergo additional processing steps such as fixation, they are highly similar to the GFP-sorted samples in read distribution (*Figure 2—figure supplement 1C—D*) and gene expression (*Figure 1G*), suggesting that these technical differences have no or minimal effects on transcriptional profiles. Taken together, our results demonstrate that that fluorescence activated nuclear sorting allows cell-type specific gene expression profiling without the requirement for transgenic animals.

## Antibody-based sorting enables the isolation and identification of subpopulations of cell types

In the FACS isolation of oligodendrocyte nuclei, we noticed that the OLIG2+ population could be subdivided based on its levels (*Figure 1—figure supplement 2B*). By flow cytometry, we found that approximately 20% of all OLIG2+ nuclei have high levels OLIG2 (High) while 80% have lower levels (Low). To determine whether these two populations form distinct subpopulations of oligodendrocytes, we isolated and analyzed them individually using RNA-seq. We found that both populations express general oligodendrocyte markers such as OLIG2, but that only OLIG2 +High nuclei express genes such as *Pdgfra* and *Cspg4* that mark oligodendrocyte precursor cells (OPCs) (*Figure 1—figure supplement 2C*). We conclude that the OLIG2+ Low population contains mature oligodendrocytes, while OLIG2+ High nuclei contain a mix of mature oligodendrocytes and immature OPCs. This is consistent with previous studies demonstrating that OLIG2 levels are higher in oligodendrocyte precursors compared to mature oligodendrocytes (*Hayashi et al., 2011*; *Kitada and Rowitch, 2006*; *Kuhlmann et al., 2008*). Furthermore, hierarchical clustering of nuclear RNA-seq profiles from all cerebellar cell types indicates that the oligodendrocyte lineage splits into two groups, mature oligodendrocytes and OPCs, with the set of all oligodendrocytes clustering with mature oligodendrocytes (*Figure 1G*). These results indicate that fluorescence activated nuclear sorting can also be used to identify subpopulations of cell types. In succeeding text and figures, these two subpopulations are analyzed separately and mature oligodendrocytes referred to simply as oligodendrocytes.

## Comparison of expression data from cell type specific populations of nuclei to single nuclei

The application of our population based, cell type specific technologies for expression profiling in the human brain provides an important complement to ongoing surveys of CNS cell types using single cell analysis. In particular, the ability to selectively isolate populations of interest allows for efficient profiling of rare cell types without the need to extensively profile common cell types. Additionally, when we compared our cerebellar glial datasets to published sequencing results from hippocampal glial single nuclei (*Habib et al., 2016*), we found that our methodology outperforms single nuclei sequencing in the number of genes detected regardless of sequencing depth (*Figure 1—figure supplement 2D-E*), and that the expression of general or cell-type specific genes is more variable across single nuclei samples than in our biological replicates (*Figure 1—figure supplement 2F*). While the variability present in single nuclei data may be desirable for addressing some biological questions, we believe that the reproducibility evidenced by highly correlated biological replicates can present important advantages for detection of cell specific molecular phenotypes associated with human pathophysiology.

## Antibody-based sorting of cell types is easily transferrable to other species

An important reason for development of this approach was to enable cell-type specific profiling in species where transgenic strains are not readily available. As a first test of this property, we chose the rat because it is a well-established model organism for behavioral neuroscience. Nuclei were isolated from rat cerebellum, fixed and stained using the same antibodies that we used in mouse, and the results were analyzed on a flow cytometer (*Figure 2A*). The staining profiles of rat nuclei resembled those of mouse (*Figure 1B*), although there were differences. For example, in the rat cerebellum, NeuN staining resulted in better separated positive and negative populations than was evident in the mouse cerebellum. As a result, additional distinct and reproducible populations of nuclei were revealed. Purification and analysis of these populations by RNA-seq revealed that some cell types could be isolated using more than one sorting strategy. For example, we found that it was possible to isolate nuclei from basket cells, astrocytes, and oligodendrocytes by staining with antibodies against SORCS3 and NeuN. This flexibility allowed us to purify all six cell types using only four types of staining (*Figure 2A*). For cell types that were isolated from more than one type of staining, we chose the population that by RNA-seq was most enriched for known markers (*) and depleted for markers of other cerebellar cell types.

To determine the group of genes that are specific to each of the rat cell types, we performed differential gene expression analysis comparing the nuclear expression profile for each cell type to unsorted cerebellar nuclei (*Figure 2B*). As expected, we observed enrichment for known markers for each cell type and depletion for markers for all other cell types. In addition, our data identified 700 to 1600 expressed genes that are significantly enriched in each of the rat cell types. We also identified genes that are most specific for each cell type using an updated version of the specificity index algorithm (*Dougherty et al., 2010*) (*Figure 2—figure supplement 1A*, *Supplementary file 3*). The advantage of this approach is that it can identify specifically expressed genes even in cell types that are very abundant in the region of interest (e.g. cerebellar granule cells). To validate both our cell-type specific rat data and the updated specificity index algorithm, we examined the expression of genes identified as being specific to each cell type using the Allen Mouse Brain Atlas in situ hybridization database (*Figure 2—figure supplement 1A*). For the top 20 most specific genes for all cell types, we observe labeling specifically in the defined cell type on average 93% of the time.

## Antibody-based sorting of cell types can be applied to postmortem human tissue

To determine whether antibody based nuclear sorting can be used productively for analysis of cell types in the human brain, we acquired postmortem human cerebellar tissue from two donors (codes XK and PK), then isolated and stained nuclei using the cell-type specific antibodies defined previously. Although the profile of stained human cerebellar nuclei is generally similar to rodent, there are differences (*Figure 2C*). For example, the conditions used to sort granule and basket neurons, astrocytes, oligodendrocytes, and OPCs were directly transferrable to human postmortem tissue,

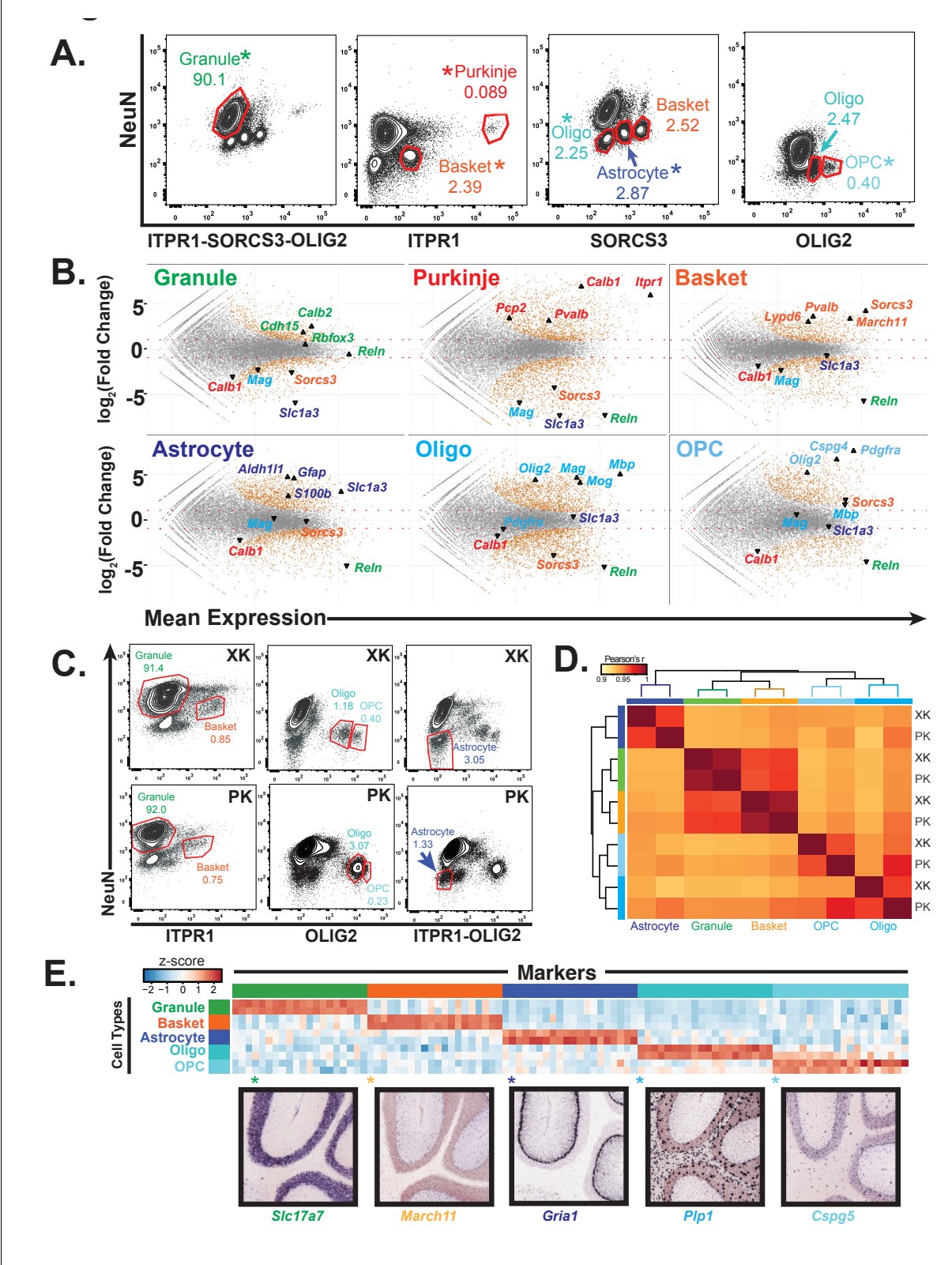

**Figure 2.** Generation of gene expression profiles for distinct cell types from rat and human cerebella. (**A**) Fluorescence activated sorting of stained nuclei from six cell types in the rat cerebellum. Antibodies are indicated on the x and y axis. When a cell type can be isolated from more than one staining scheme, the population used for downstream analysis is indicated with (*). Percentage of population in each gate is indicated. (**B**) Differential expression analysis of antibody sorted nuclei for six cell types compared to unsorted nuclei from the rat cerebellum. Known markers for each cell type

*Figure 2 continued on next page*

*Figure 2 continued*

are highlighted: granule (*Cdh15*, *Calb2*, *Rbfox3*, *Reln*), Purkinje (*Pcp2*, *Pvalb*, *Cabl1*, *Itpr1*), Basket (*Lypd6*, *Pvalb*, *Kit*, *Sorcs3*), Astrocyte (*Aldh1l1*, *Gfap*, *S110b*, *Slc1a3*), all oligodendrocytes (*Olig2*), mature oligodendrocyte, (Labeled Oligo: *Mag*, *Mog*, *Mbp*), oligodendrocyte precursor cells, (Labeled OPC: *Cspg4*, *Pdgfra*). (C) Fluorescence activated sorting of granule, basket, astrocyte, mature oligodendrocyte, and OPC nuclei from the cerebellum of two human samples (XK and PK). Percentage of population in each gate is indicated. (D) Heatmap showing Pearson's correlation coefficient between human samples. Hierarchical clustering is performed using the 250 most variable genes across samples. (E) Heatmap showing the 20 most specific genes for each human cell type as identified by the Specificity Index algorithm. Rows: sorted nuclei from XK, PK samples; columns: genes enriched in each cell type. Color represents the z-score of gene expression compared to all samples. Lower panels show examples of gene expression from the Allen Mouse Brain Atlas for genes identified as highly specific based on the Specificity Index analysis of human cell types. Allen Brain Atlas example gene is with (*) on heatmap. See also *Figure 2—figure supplement 1*.

DOI: https://doi.org/10.7554/eLife.37551.005
The following figure supplement is available for figure 2:

**Figure supplement 1.** Analysis of cell type specific gene expression profiles generated from rat and human cerebella identifies known mouse marker genes for each cell type, related to *Figure 2*.
DOI: https://doi.org/10.7554/eLife.37551.006

whereas those used to sort mouse and rat Purkinje cell nuclei were not successful for isolation of human Purkinje cell nuclei, possibly because while Purkinje cells are already rare in mouse and rat (0.1% in mouse, 0.05% in rat; *Figures 1B* and *2A*), they have been reported to be even more rare in human (*Korbo and Andersen, 1995*). We also observed slight differences in staining between the two postmortem human samples: in XK the oligodendrocyte and OPC populations are well separated, whereas they are more difficult to distinguish in the sample PK. To assess whether these differences might prevent cell type specific analysis of human postmortem samples, we profiled gene expression in nuclei isolated from five cell types (granule, basket, astrocyte, oligodendrocyte, OPC) in these two individual brains using RNA-seq. Examination of known markers from each of these cell types established that relevant cell specific markers are enriched in the data for each cell type, and depleted from the other cell types (*Figure 2—figure supplement 1B*). Calculation of pairwise Pearson correlation coefficients across the ten datasets revealed that the two neuronal cell types (granule and basket) were highly correlated between samples (r = 0.99 for both), while the glial data had slightly lower correlations (r = 0.96–0.97). Hierarchical clustering resulted in separation of the samples by cell type, indicating that the variation across cell types is stronger than that between individual samples (*Figure 2D*). Finally, we applied the specificity index algorithm to these data to identify specifically expressed genes in each human cell type (*Figure 2E*). When we examined the genes that are most specific, we noted that many of these genes were also highly expressed in the same cell type in mouse and rat, but some genes appeared to be human-specific in each cell type (*Supplementary file 3*).

## Cross-species comparison reveals cell-type and species specific genes

Given the high quality of the cell specific expression profiles that we generated for mouse, rat, and human, we were interested in addressing directly the extent of gene expression conservation and divergence between species in these well-defined, classical cerebellar cell types. To ensure that our results reflected altered expression rather than differences in annotation between species, we limited our analysis to high-confidence 1:1 orthologous genes between mouse, rat and human (*Figure 3—figure supplement 1A*, methods). To explore the similarities and differences across species, we used hierarchical clustering to determine the relationships between samples. Interestingly, we found that when we performed clustering using all genes, the samples resolved primarily by species (*Figure 3—figure supplement 1B*). However, when we performed clustering using the 250 most variable genes across samples, they clustered primarily by the cell type (*Figure 3A*). To further investigate this result, we computed the pairwise Pearson correlation coefficients across all samples (*Figure 3—figure supplement 1C*). We found that the range of correlation coefficients between the samples for the same cell type across species is similar to that for cell types isolated from the same species, suggesting that both cell type and species are important determinants of the expression profiles. Our interpretation of these data is the biggest differences (most variable genes) define cell types, while the accumulation of more subtle differences define species. To explore whether species also contributes to bigger differences, we performed principal components analysis using the 250

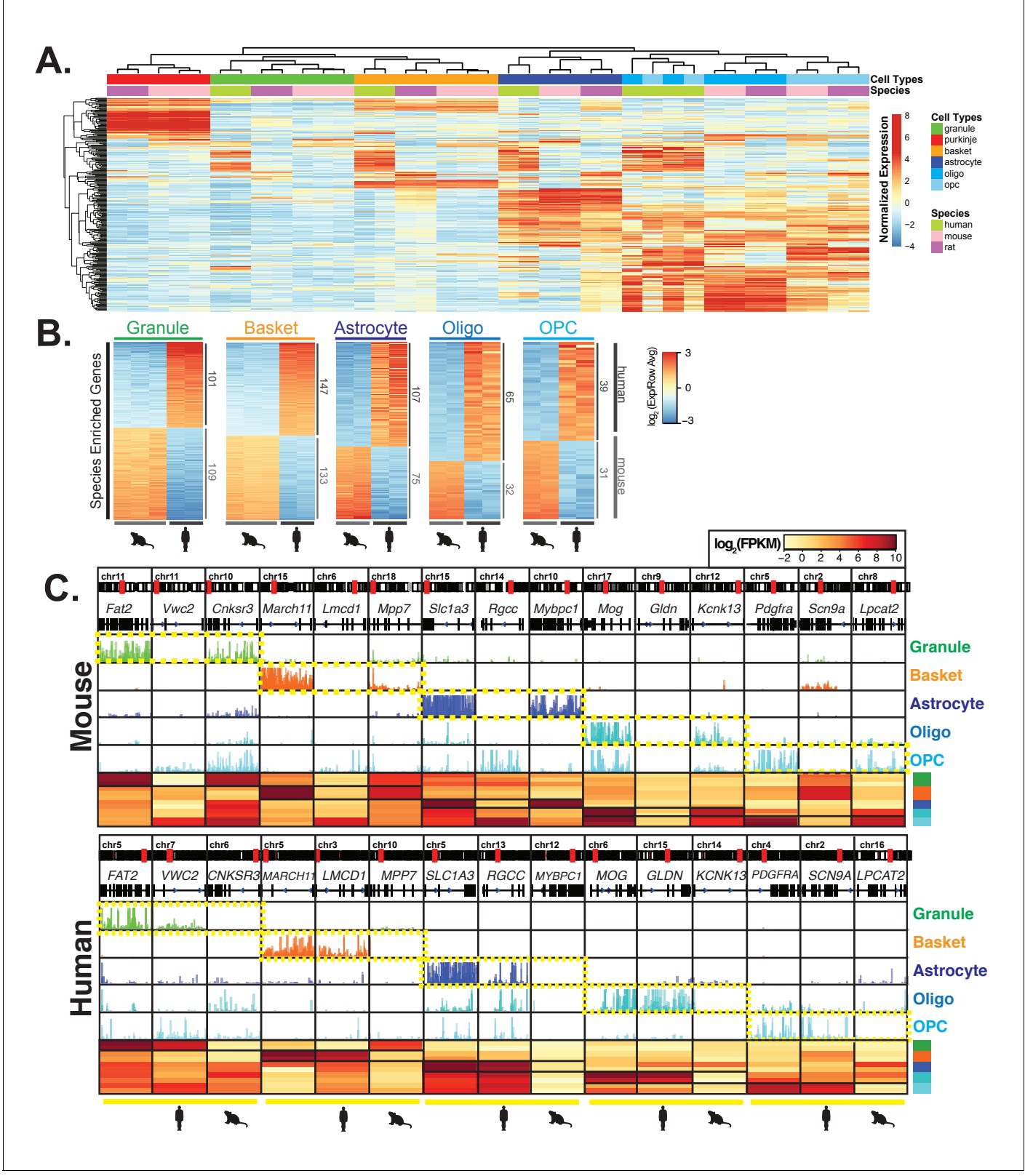

**Figure 3.** Comparative analysis of gene expression across species reveals cell-type and species specific differences. (**A**) Heatmap showing normalized expression of the 250 most variable genes across all samples. Hierarchical clustering is performed using these genes and reveals that samples cluster primarily by cell type and secondarily by species. (**B**) Heatmap showing mouse- and human-enriched genes for each cell type, excluding any genes that are mouse- or human-enriched across all cell types. The number of genes that are significantly human or mouse enriched is indicated. (**C**) Browser view

*Figure 3 continued on next page*

*Figure 3 continued*

and heatmaps showing for each cell type, an example of a shared marker gene, a human-enriched gene, and a mouse-enriched gene. See also *Figure 3—figure supplement 1*, *Figure 3—figure supplement 2*, *Figure 3—figure supplement 3*.

DOI: https://doi.org/10.7554/eLife.37551.007

The following figure supplements are available for figure 3:

**Figure supplement 1.** Comparative analysis of gene expression in specific cell types of mouse, rat, and human, related to *Figure 3*.

DOI: https://doi.org/10.7554/eLife.37551.008

**Figure supplement 2.** Detailed comparative analysis of gene expression in five cerebellar cell types in mouse and human, related to *Figure 3*.

DOI: https://doi.org/10.7554/eLife.37551.009

**Figure supplement 3.** Expression of species-enriched genes in published cerebellar single nuclei/cell RNA-seq data, related to *Figure 3*.

DOI: https://doi.org/10.7554/eLife.37551.010

most variable genes across samples (*Figure 3—figure supplement 1D-F*). Examination of the first eight principal components (PC1-8), which collectively account for 90.5% of the variability across all samples, revealed that PC1 and 3 categorize samples by the cell type, while PC2, 6, and 8 split samples by species. The other principal components organize samples by both cell type and species. Because PC1 and 3 account for 47% of the variability across samples while PC2, 6, and 8 account for only 22% of variability (*Figure 3—figure supplement 1D*), we conclude that the most important differences between the profiles result from the cell-type specific expressed genes that are shared among species, but that some important differences also arise due to the expression of species-specific genes.

It is apparent from these data that genome-wide quantitative differences in expression profiles between species must also be considered when assessing the fine-tuned functional properties of a given cell type in different species. We note that while we have tried to minimize the effect of annotation differences between species by limiting our analyses to high-confidence 1:1 orthologous genes, it is possible that some of the species differences that we observe can be attributed to differences in annotation. This may explain why hierarchical clustering using all genes results in clustering by species while clustering with the most variable genes results in clustering by cell type. To gain further insight into cell-type specific functions that have changed across species, we calculated the specificity index for all genes in a given species. We next compared the rank of the 100 most specific genes for each cell type in mouse with their ranks in rat or human (*Figure 3—figure supplement 1G–H*, *Supplementary file 3*). We found that, in general, genes that are highly specific in mouse are also highly specific in rat and human but there are many exceptions. As expected, the specificity rank was more conserved between mouse and rat than mouse and human.

To explore further the extent to which the expression of cell-type specific genes change across species, we focused our comparative analysis on the data from mouse and human because the annotations of these genomes are more complete than the annotations for the rat genome. As a result, the number of high-confidence orthologs that could be used for comparison increased from 11,443 to 14,273 (*Figure 3—figure supplement 2A*). To focus on differentially expressed genes that are most likely to impact cellular function, we selected only genes that passed a stringent adjusted p-value cutoff of <10e-5, that changed in expression by at least 4-fold between species, and that are expressed at significant levels in each cell type (average number of normalized counts (DEseq2 base-Mean)>400 and average $\log_2$(fpkm) in at the species with higher expression >4). Using these criteria, we found between 132 and 408 genes that are mouse or human enriched in these cell types (*Figure 3—figure supplement 2B*). To identify cell-type and species differences rather than general differences between mouse and human, we removed from the analysis genes that are differentially expressed between mouse and human in all cell types analyzed (adjusted p<10e-5, fold change >2, no expression cutoff). Filtering in this way yielded between 70 and 280 cell-type and species specific differentially expressed genes that were approximately equally distributed between those enriched in mouse and those enriched in human (*Figure 3B*, S5B, *Supplementary file 4*). Examples of shared and species-specific expressed genes are shown in *Figure 3C*. Interestingly, expression of these genes in rat most often resembles expression in mouse (*Figure 3—figure supplement 2C*).

Because our analysis compares the expression of populations of nuclei, it is possible that some of the species enriched genes that we have identified could be the result of sorting for slightly different subpopulations in mouse and human rather than real species differences. To test this, we examined

the expression of our species-enriched genes in published human cerebellar single nuclei RNA-seq data from (*Lake et al., 2018*) and mouse cerebellar single cell RNA-seq data from (*Saunders et al., 2018*). We noted that although Lake et al. label two of their clusters as populations of Purkinje neurons, these clusters actually contain cerebellar GABAergic interneurons (no cerebellar interneurons were identified in the publication), as both clusters express cerebellar interneuron markers such as *SLC6A1 and TFAP2B* (*Chiu et al., 2002*; *Zainolabidin et al., 2017*), and neither express well characterized Purkinje markers such as *CALB1* and *CA8* (*Jiao et al., 2005*; *Taniuchi et al., 2002*; *Tolosa de Talamoni et al., 1993*; *Whitney et al., 2008*) (*Figure 3—figure supplement 3A* - left). Instead, the group labeled Purk1 express basket cell markers *SORCS3* and *LYPD6* (*Doyle et al., 2008*; *Hermey et al., 2004*; *Schilling and Oberdick, 2009*)(*Figure 3—figure supplement 3A* - left). These marker genes for Purkinje neurons, GABAergic interneurons, and basket neurons are correctly expressed in the mouse data from Saunders *et al.*, although we note that while basket neuron subtypes 3 and 4 express *Slc6a1* and *Tfap2b*, they do not express *Sorcs3* and *Lypd6* (*Figure 3—figure supplement 3A* - right). Thus, we have renamed the two populations from Lake *et al.* Basket and Interneuron type 2 (Int2). In keeping with the conventions of this paper, we also renamed their Cerebellar astrocyte population Bergmann glia.

To assess whether our species-enriched genes are heterogeneously expressed in single nuclei/cell data, we first examined violin plots of gene expression of example marker, human-enriched, or mouse-enriched genes (*Figure 3—figure supplement 3B*). In human single nuclei data, expression of the human-enriched granule cell genes *VWC2* and *CCDC175* revealed granule cell-specific expression similar to granule marker genes *FAT2* and *RBFOX3*. However, mouse-enriched granule cell genes *CNKSR3* and *ECE1* displayed only background levels of expression across all cell types (*Figure 3—figure supplement 3B* - left). In contrast, in mouse single cell data, expression of the mouse-enriched granule cell genes *Cnksr3* and *Ece1* revealed granule cell expression similar to granule marker genes *Fat2* and *Rbfox3*, while human-enriched granule cell genes *Vwc2* and *Ccdc175* displayed only background levels of expression across all cell types (*Figure 3—figure supplement 3B* - right).

To quantify these observations for all cell-type specific species-enriched genes, we used two metrics to measure gene expression in single nuclei/cell populations - mean expression levels and the proportion of nuclei with expression over a threshold one $\log_2$(TPM). For each cell type, we measured expression using these two metrics for all human- or mouse-enriched genes in the cluster (s) corresponding to that cell type. For example, for granule cells, we quantified mean expression and proportion of nuclei with expression for all 101 human-enriched or 109 mouse-enriched genes. As a reference, we also quantified expression for 472 housekeeping genes that are unchanged in expression between mouse and human. For astrocytes, because our sorted populations capture both conventional astrocytes and Bergmann glia (specialized cerebellar astrocytes), we examined the expression of our 107 human-enriched or 75 mouse-enriched genes in the single nuclei populations for both astrocytes and Bergmann glia. Saunders *et al.* subdivide their basket neurons into four subpopulations, non-Bergmann glia astrocytes into two subpopulations, and oligodendrocytes into two subpopulations; we also examined the expression of our human and mouse-enriched genes for these cell types in all subpopulations. We found that for all cell types and for both measures of expression, the expression of human-enriched genes in human single nuclei data is significantly higher than the expression of mouse-enriched genes (*Figure 3—figure supplement 3C*). Concordantly, the expression of mouse-enriched genes in mouse single cell data is higher than the expression of human-enriched genes for all cell types, including subpopulations of each cell type (*Figure 3—figure supplement 3D*). For both measures of expression, the distributions of expression for mouse- and human-enriched genes are statistically different for all cell types except for OPCs, which is likely due to sample size as Saunders *et al.* only identified eight cerebellar OPCs. These results confirm that the genes we have identified as differentially expressed between human and mouse are likely to be true species differences, and do not arise as a result of sorting different subpopulations in the two species.

We next performed Gene Ontology (GO) analysis of species-specific genes to understand whether their expression might reflect changes that occur independently, or arise due to transcriptional programs that are altered between species. GO analysis of human-enriched genes from granule neurons, astrocytes, oligodendrocytes, and OPCs do not have significantly associated GO categories, suggesting species differences in these cell types contribute to a wide range of pathways

and functions. The majority of human-enriched genes in basket cells are also unrelated functionally, although there was significant enrichment for the cellular component sub-ontology cell projection membrane. GO analysis of mouse-enriched genes from astrocytes revealed weak enrichment for genes with phosphoric diester hydrolase/phospholipase activity, while mouse-enriched genes from granule neurons, basket neurons, oligodendrocytes, and OPCs revealed enrichment for genes with transporter activity or those involved in neuronal development (*Supplementary file 9*). These data indicate that evolutionary divergence between mouse and human cell types involves both frequent alterations in regulatory mechanisms impacting single genes (*Villar et al., 2015*), and those regulating conserved, functionally important pathways (*Nord et al., 2015*; *Rebeiz et al., 2015*).

To confirm the presence or absence of some of these species and cell-type specific genes (*Figure 4—figure supplement 1A-B*) in the mouse brain, we examined publicly available gene expression data from the Allen Mouse Brain Atlas and GENSAT Project (*Gong et al., 2003*; *Lein et al., 2007*) (*Figure 4—figure supplement 1C*). These data demonstrate expression in the granule layer of the cerebellum for the mouse-enriched genes *Cnksr3*, *Ece1*, and *Pde1c*, and the absence of expression of the human-enriched genes *Ccdc175*, *Clvs2*, *Vwc2*, and *Pde1a* (*Figure 4—figure supplement 1C*). Interestingly, while *Clvs2* and *Pde1a* are absent from the cerebellum, they are expressed in other regions in the mouse brain, suggesting that alterations in cell specific regulatory sequences may underlie these species differences (*Carroll, 2005*; *Carroll, 2008*).

## Enhanced chromatin accessibility of species specific expressed genes

As an independent measure of actively expressed genes, we examined chromatin accessibility in mouse and human nuclei using Assay for Transposase-Accessible Chromatin using sequencing (ATAC-seq). Since our sorted nuclei are fixed, we used the modified ATAC-seq protocol for fixed nuclei from (*Chen et al., 2016*); in that publication, the authors demonstrate that fixation does not affect tagmentation efficiency and that data generated from fixed nuclei are highly correlated with those from unfixed nuclei (R = 0.93). Since previous studies have demonstrated that ATAC peaks are typically found in the promoters and gene bodies of expressed genes and severely reduced or absent from genes that are not expressed (*Buenrostro et al., 2013*; *Mo et al., 2015*; *Su et al., 2017*), we collected ATAC-seq data from mouse and human cerebellar granule and basket nuclei to confirm the expression data. The majority of ATAC peaks in our samples are located in promoters or gene bodies (*Figure 4—figure supplement 2A*). As expected, protein coding genes that are identified as expressed in our data have ATAC peaks in their promoters and gene bodies.

The locus containing *JKAMP* [1], *CCDC175* [2], and *RTN1* [3] provides an example of the categories of gene expression changes that are evident in our data. In human granule cells, all three genes are expressed at significant levels, as is evident by the presence of primary transcripts over the gene body, and ATAC peaks at the promoter (*Figure 4A* - top). In mouse granule cells, expression of the *Rtn1* gene [1] is conserved, with similar levels of nuclear transcripts (7.6 $\log_2$(normalized counts) in human vs 8.1 in mouse) and promoter ATAC coverage (2.6 $\log_2$(FPKM) in mouse vs 2.9 in human). In to contrast the robust expression of the *CCDC175* gene [2] evident in human granule cells, no primary transcripts and no promoter ATAC peaks are observed for the mouse gene (we did not quantify expression or ATAC coverage for CCDC175 because the mouse and human genes are annotated as low confidence orthologs and thus excluded from quantitative analysis). This gene, therefore, is an example of a human specific granule cell expressed gene. The third gene in this locus *Jkamp* [1], is an intermediate case. Although *Jkamp* is expressed in both human and mouse granule neurons, its expression in mouse is reduced approximately five-fold (8.6 $\log_2$(normalized counts) in human vs 6.3 in mouse) and the accessibility of its promoter is reduced (ATAC coverage 3.8 $\log_2$(FPKM) in human vs 1.7 in mouse). Within this genomic locus, therefore, examples of conserved, species specific and quantitatively variable expression are evident for adjacent genes in a single homologous cell type.

To validate that our ATAC data are cell type specific and relate to gene expression, we first performed differential expression analysis to identify granule- and basket- specific cells in mouse and human. We identified 1100 granule- and 1151 basket-enriched genes in human, and 1313 granule- and 1124 basket-enriched genes in mouse. We next analyzed ATAC reads over the promoters of these cell-type specific genes. As expected, we found that in both human and mouse granule neurons, the promoters of granule-enriched genes have stronger ATAC signal compared to the promoters of basket-enriched genes (*Figure 4—figure supplement 2B-C*). The reverse is true for

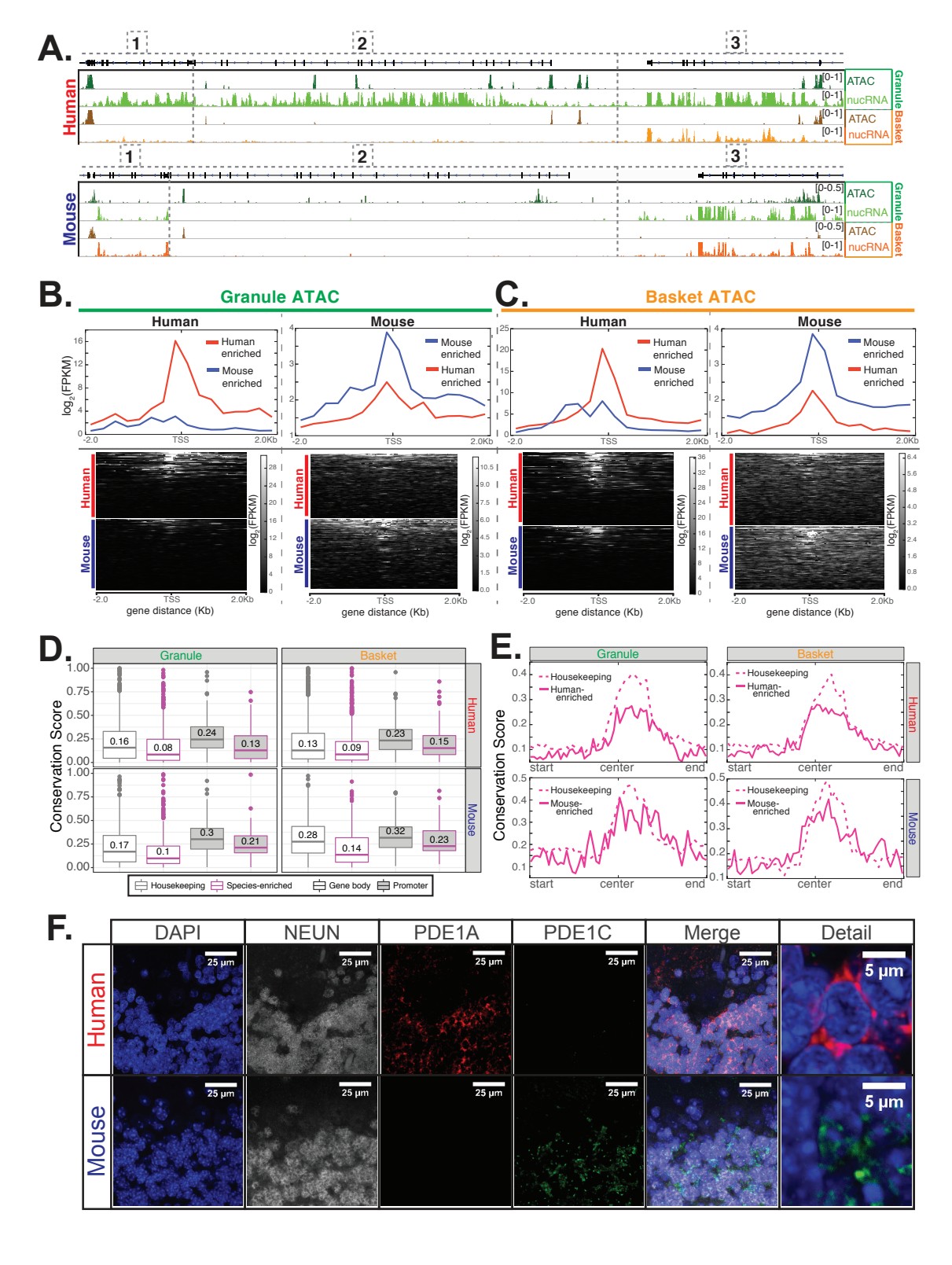

**Figure 4.** Epigenetic and immunofluorescence validation of gene expression differences between mouse and human cerebellar granule cells. (A) Browser views showing a homologous region of approximately 150 kb from chr14 in human and chr12 in mouse. Minimum and maximum data range values are indicated for each track. Genes located in this region are: [1] *JKAMP* (human)/*Jkamp* (mouse) [2] *CCDC175/Ccdc175* [3] *RTN1/Rtn1*. For each species, four tracks are shown: ATAC-seq DNA accessibility from granule nuclei (dark green), nuclear RNA levels from granule nuclei (green), ATAC-seq

*Figure 4 continued on next page*

*Figure 4 continued*

from basket nuclei (dark orange), nuclear RNA levels from basket nuclei (orange). The merged profile of two biological replicates is shown for all tracks. All three genes in the locus are strongly expressed in human granule cells and are associated with the presence of ATAC DNA accessibility sites. *JKAMP/Jkamp* and *RTN1/Rtn1* are also expressed in human basket cells and mouse granule and basket cells and are also associated with DNA accessibility peaks. *CCDC175/Ccdc175* is not expressed in human basket cells or mouse granule or basket cells; correspondingly, the promoter and gene body of this gene are depleted for ATAC DNA accessibility sites in these three cell types compared to human granule cells. (B-C) Analysis of ATAC-seq DNA accessibility assay from human (left) or mouse (right) from sorted cerebellar granule (B) or basket (C) cell nuclei. (B) Top: metagene analysis showing the median log$_2$(FPKM) of reads from the promoter regions of 101 human-enriched or 109 mouse-enriched granule cell species specific genes. Bottom: read density of ATAC-seq reads over the promoter of each gene individually. (C) Top: metagene analysis showing the median log$_2$(FPKM) of ATAC-seq reads from the promoter regions of 147 human-enriched for 133 mouse-enriched basket cell species specific genes. Bottom: read density of ATAC peaks over the promoters of each gene individually. (D-E) Analysis of DNA sequence conservation for accessible chromatin regions as defined by ATAC-seq peaks. (D) Analysis of DNA sequence conservation in promoter (magenta) or gene body (grey) ATAC peaks. Top: boxplots of 100-way human PhastCons scores for human cerebellar granule (left) or basket (right) cell ATAC peaks that are associated with 101 human-enriched granule cell genes (magenta outline, left), 147 human-enriched basket cell genes (magenta outline, right), or 472 genes that are not significantly differentially expressed between mouse and human (housekeeping, grey outline, left and right). Bottom: boxplots of 60-way mouse PhastCons scores for mouse cerebellar granule (left) or basket (right) cell ATAC-seq defined peaks that are associated with 109 mouse-enriched granule cell genes (magenta outline, left), 133 mouse-enriched basket cell genes (magenta outline, right), or 472 genes that are not significantly differentially expressed between mouse and human (housekeeping, grey outline, left and right). Promoter peaks are filled in grey; gene body peaks are unfilled. Median values for each group are indicated. T-test (one-sided, unequal variance) p-values comparing conservation scores for species-specific versus housekeeping genes: human granule promoter (0.0047), human granule gene body (1.0e-6), human basket promoter (0.017), human basket gene body (1.1e-14), mouse granule promoter (0.13), mouse granule gene body (2.8e-8), mouse basket promoter (0.25), mouse basket gene body(1.4e-8). (E) Metagene analysis showing mean 100-way human PhastCons scores (top) or 60-way mouse PhastCons scores (bottom) across the length of ATAC-seq peaks located in the promoters of species-enriched genes (solid magenta lines) or housekeeping genes (dashed magenta lines). (F) Immunofluoresence confirming the expression of PDE1A and PDE1C in mouse and human cerebellar slices. In mouse, PDE1C (green) is present specifically in granule cells. PDE1A (red) is not expressed above the background of the assay. In human cerebellum, PDE1A (red) is evident specifically in granule cells, and background labeling is observed for PDE1C (green). NeuN (grey) is a marker for granule cells. Staining was performed at least two times each using sections from two separate mice and two human donors. Images shown are representative of all data collected. See also *Figure 4—figure supplements 1–2*.

DOI: https://doi.org/10.7554/eLife.37551.011

The following figure supplements are available for figure 4:

**Figure supplement 1.** Analysis of cell-type specific ATAC peaks and examples of mouse- and human-enriched genes in granule cells, related to *Figure 4*.

DOI: https://doi.org/10.7554/eLife.37551.012

**Figure supplement 2.** Relationship between chromatin accessibility and gene expression in cerebellar granule and basket neurons, related to *Figure 4*.

DOI: https://doi.org/10.7554/eLife.37551.013

basket cells as the promoters of basket-enriched genes have elevated ATAC signal compared to the promoters of granule-enriched genes. To further validate the relationship between chromatin accessibility and gene expression, we identified differentially accessible regions between granule and basket neurons in both human and mouse (*Supplementary file 5*), and analyzed the expression of genes associated with these regions (*Figure 4—figure supplement 2D*). We found that in general, the presence of accessible regions in the 5' end of genes is associated with elevated gene expression. For example, peaks that are enriched in human granule neurons are also associated with elevated gene expression in granule neurons when these peaks are located in promoter, 5'UTR, exon, or intron regions, but not when these peaks are located in 3'UTR or TTS regions (*Figure 4—figure supplement 2D*).

We next analyzed granule nuclei ATAC reads over the promoters of the 101 human-enriched and 109 mouse-enriched granule cell specific genes that we identified by differential gene expression analysis. We found that in human, the majority of the 101 human-enriched genes have strong ATAC signal over the promoter region, while only a few of the 109 mouse-enriched genes have strong promoter ATAC signals (*Figure 4B* – left). In mouse, the reverse is true – most of 109 mouse-enriched genes have strong promoter ATAC signals compared to only a few of the 101 human-enriched genes (*Figure 4B* – right). We obtained similar results from the ATAC reads from basket cells. In human, the majority of the 147 human-enriched genes have strong ATAC signal over the promoter region, while only a few of the 133 mouse-enriched genes have strong ATAC signals (*Figure 4C* – left); in mouse, the reverse is true (*Figure 4C* – right). Since it has been demonstrated repeatedly

that the regulatory regions of expressed genes have accessible chromatin structures (*Cockerill, 2011*; *Gross and Garrard, 1988*; *Weintraub and Groudine, 1976*), these data provide independent confirmation of the species differences detected by the gene expression analysis.

To determine whether the differences in chromatin accessibility between mouse and human might be explained by evolutionary changes in DNA sequence, we examined whether DNA bases contained within ATAC peaks associated with mouse- or human-enriched genes are less conserved when compared to genes that are unchanged in expression between mouse and human. To measure sequence conservation, we used PhastCons scores, which estimate from multiple species sequence alignments the likelihood that a base is under purifying selection. As a reference, we identified 472 well-expressed genes that are not significantly differentially expressed between mouse and human (p>0.2 using samples from all cell types, DESeq2 baseMean >400, $\log_2$(fpkm) >4), which we call housekeeping genes. We found that DNA bases contained within ATAC peaks associated with mouse- or human-enriched genes are less conserved than those associated with housekeeping genes (*Figure 4D*, left for granule cells, right for basket cells). Analysis of conservation scores over the normalized length of promoter ATAC peaks reveals that while the centers of defined peaks are more conserved compared to the beginning or end of peaks, this difference is more prominent for housekeeping genes that for human- or mouse-enriched genes (*Figure 4E*, left for granule cells, right for basket cells). These results suggest that differences we observe in the expression of cell-type and species-specific genes between human and mouse might result from evolutionary driven differences in promoter chromatin accessibility between the two species.

While examining the genes that differ in granule cells between mouse and human, we noted an interesting example that involves changing expression of alternative family members. The *Pde1* gene family contains the three calcium- and calmodulin-dependent phosphodiesterases *Pde1a*, *Pde1b*, and *Pde1c*. These enzymes catalyze the hydrolysis of cAMP or cGMP, and they play important roles in modulation of neuronal activity. Our data demonstrate that mouse granule cells express only *Pde1c*, whereas granule cells from human express high levels of *PDE1A* and low levels of *PDE1C* (*Figure 4—figure supplement 1B*). Our results match previous findings that *PDE1A* is expressed in the human but not mouse cerebellum (*Loughney and Ferguson, 1996*; *Loughney et al., 1996*; *Sonnenburg et al., 1993*). To further confirm this result, we assayed levels of PDE1A and PDE1C in mouse and human cerebellar sections by immunofluorescence (*Figure 4F*). We found, consistent with the RNA-seq and ATAC data, that both PDE1 members are primarily localized in the granule layer, with PDE1C expression predominant in mouse and PDE1A expression predominant in human cerebellum (*Figure 4F*). Since PDE1A preferentially hydrolyzes cGMP while PDE1C hydrolyzes both cAMP and cGMP with equal efficiencies (*Takimoto, 2009*), these differences in gene expression may result in changes in neuronal activity between granule cells from the two species.

To assess whether some of these species differences may have arisen due to changes in *cis*-regulatory regions, we identified motifs that are enriched in accessible regions associated with mouse or human enriched genes (*Supplementary file 6*). In general, we found that only a few motifs reach significance after correcting for multiple testing, and these motifs tend to correspond to common transcription factors such as CTCF or cell-type specific transcription factors such as NEUROD1 for granule cells. However, we noticed that the top motifs for gene body peaks in mouse-enriched granule cell genes contain motifs for 4 GATA transcription factors (GATA2, GATA6, GATA1, and GATA4). Although none of these motifs reached significance after correcting for multiple hypothesis testing (Benjamini q-values: 0.0676, 0.1069, 0.176, and 0.2192), we were intrigued by this result as none of the conventional GATA family members (*Gata1 - 6*) are expressed in mouse granule cells (*Figure 4—figure supplement 1D*). However, we found that *Trps1*, which encodes an atypical GATA factor that has been shown to specifically bind to the consensus GATA sequence (*Malik et al., 2001*), is expressed in granule cells in mouse but not human (*Figure 4—figure supplement 1D*). A previous study has found that while human mutations in the GATA binding domain of *TRPS1* cause defects in bone development, these phenotypes are not completely recapitulated in the mouse model (*Malik et al., 2002*). Although there are no reports of neurological phenotypes in the mosue model, our results suggest that changes in the pattern of transcription factor expression between mouse and human may contribute to discrepancies in phenotypes between mouse models and human disease.

Taken together, these data demonstrate that nuclear RNA-seq data can be employed to accurately assess cell-type specific gene expression in rodent and human brain, and they allow several important conclusions regarding gene expression in the nervous system. First, gene expression in the brain is highly conserved in specific cell types across species. Accordingly, most abundant and cell-specific markers of well characterized cell types are shared between rodent and human brain. Second, despite this shared identity, there are a significant number of genes in each cell type whose expression is not conserved between species. This group of genes does not generally conform to known GO categories, and in some cases, they are expressed in other brain regions. Third, cell-type and species-specific expression can reflect altered regulation of adjacent genes within a locus, or selective expression of functionally related yet distinct members of a given gene family. Finally, cell-type specific changes in the expression of transcription factors may impact the expression of downstream genes. Although the consequences of these gene expression changes will have to be interrogated in future studies, our data demonstrate that important differences in gene expression in specific CNS cell types occur between species and suggest that they may result in functionally important differences in the biochemical functions of even the most classically defined rodent and human cell types.

## Identification of clinically relevant SNPs that are located in putative cell-type specific gene regulatory regions

To assist in interpreting the functional relevance of non-coding SNPs from GWAS studies, we overlapped the coordinates of clinically relevant SNPs from the NHGRI-EBI GWAS catalog with the cell-type specific regulatory regions identified from our human ATAC-seq data (*Supplementary file 7*). We found 767 and 1381 SNPs that are located in accessible regions associated with granule and basket neurons respectively. Of these, 163 and 363 SNPs are located in regions that are differentially accessible in granule and basket neurons. Although further work is required to confirm that these SNPs are functionally significant, these results may provide insight into the mechanism by which SNPs contribute to human disease. For example, while many SNPs in *SNCA*, the gene that encodes alpha-synuclein, have been strongly associated with both familial and sporadic Parkinson's disease (*Campêlo and Silva, 2017*; *Stefanis, 2012*), less is known about the mechanism by which non-coding SNPs in *SNCA* contribute to disease risk. We found that *SNCA* is expressed at higher levels in granule neurons relative to basket neurons (*Figure 4—figure supplement 1E*); correspondingly, we identified seven regions around the *SNCA* gene that are preferentially accessible in granule neurons (*Figure 4—figure supplement 1E*) and may regulate *SNCA* expression in *cis*. These putative regulatory regions contain two SNPs that have previously been linked to Parkinson's - rs356182 and rs2736990. Interestingly, the GG genotype of rs356182, which has been associated with a tremor-dominant, slower-progressing form of Parkinson's, is also associated with lower levels of *SNCA* expression in the cerebellum but not in any other brain regions including substantia nigra, caudate, and frontal cortex (*Cooper et al., 2017*). Because a major challenge of these studies is in determining whether identified SNPs are themselves functionally significant or whether they are merely linked to a functional SNP, the ability to determine which SNPs reside in cell-type specific regulatory regions may enable the generation of testable hypotheses for how GWAS derived SNPs contribute to human diseases.

## Nuclear profiling of three cell types isolated from 16 postmortem human brains

Given the genetic heterogeneity of human populations, the variations in human tissue processing times, and the difficulties cited in previous studies of total RNAs isolated from human tissue (*McCall et al., 2016*; *Webster, 2006*), we were interested in determining sources of variability across samples from different individuals. Since differences in cellular composition can confound analyses at the tissue level (*Jaffe and Irizarry, 2014*), we applied our methodology to analyze cell-type specific gene expression in an additional 14 human brain samples. In total, we obtained samples from 16 control donors, roughly equally split between genders (seven males and nine females), and across ages (four individuals each in the following age groups: 20–30, 40–50, 60–70, 80–90 years) (*Figures 2C* and *5A*, *Supplementary file 2*). We prepared nuclei from all 16 samples and stained them using antibodies against ITPR1 and NeuN – a staining strategy that allowed us to purify

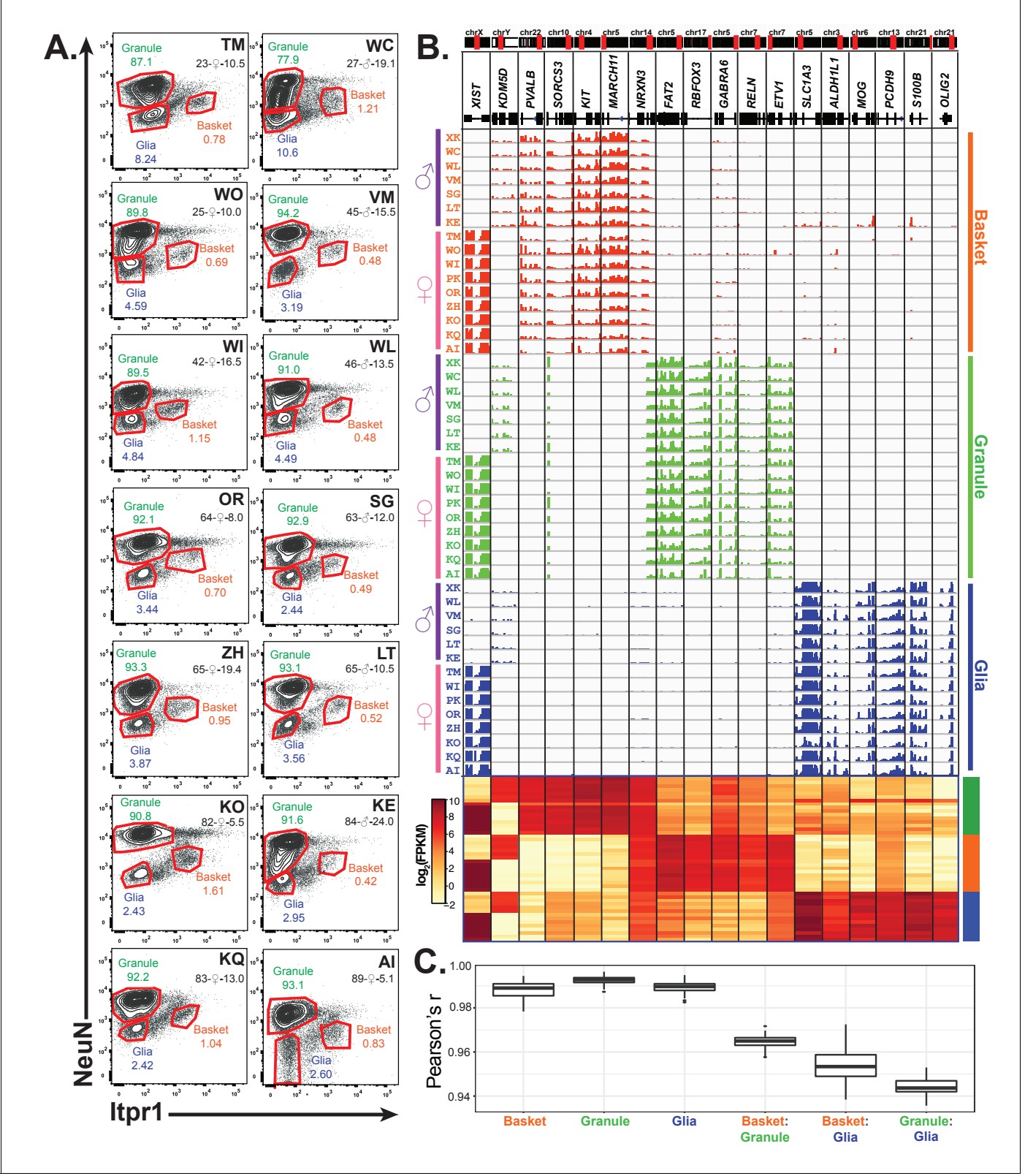

**Figure 5.** Profiling of three cell types in 16 human cerebellar samples. (**A**) Fluorescence activated nuclear sorting of three cell types from the cerebella of 14 individuals. The percentage of each cell type in different individuals is shown. Each sample is identified by a two-letter code. Also shown are the age, gender, and post-mortem delay interval for each sample. (**B**) Browser view and heatmaps showing gene expression for gender and cell-type

*Figure 5 continued on next page*

*Figure 5 continued*

markers across 16 individuals (14 from A and two from *Figure 4*). For glia, samples from WC and WO were excluded due to granule cell contamination. (C) Boxplot showing pairwise Pearson's correlation coefficient within cell types and between cell types for different individuals.
DOI: https://doi.org/10.7554/eLife.37551.014

in a single sort nuclei from three different cell types: granule cells, basket cells, and total glia (*Figures 2C* and *5A*). We noticed that, although the staining pattern varied slightly between samples, we could easily distinguish the ITPR1- NeuN+ population containing granule cells and the ITPR1- NeuN- population containing total glia. The only exceptions were the glial populations from samples WC and WO in which the ITPR1- NeuN- population did not separate well from the ITPR1- NeuN+ population (*Figure 5A*). Gene expression analysis of these two samples revealed that these two sorts failed to effectively separate granule cells from glia, and they were excluded from further analysis. Although the population containing basket cells appeared to be highly variable across individuals, for most of the samples, we could easily identify and gate for a distinct ITPR1+ NeuN population that was well separated from other nuclei. In some individuals (e.g. WC and KE) where this population was less distinct, we gated based on where this population appeared in other samples while taking care to avoid auto-fluorescent ITPR1+ NeuN+ and glial ITPR1- NeuN- nuclei. Consequently, our analysis included 46 human datasets: 16 from granule cells, 16 from basket cells, and 14 from glia. As shown in *Figure 5B*, RNA-seq analysis of these datasets demonstrate that the isolated populations are highly specific, as each sample is enriched for the appropriate markers for that cell type, and depleted for markers of other cerebellar cell types. As expected, only samples from females express the X-inactive specific transcript (*XIST*), while only male samples express Y chromosome genes such as *KDM5D*. Pairwise Pearson correlation coefficients between biological replicates for each cell type are between 0.98 and 1 (*Figure 5C*). These data demonstrate that, despite differences between individuals in the abundance of each cell type or the recovery of nuclei during preparation and sorting, the cell-type specific human expression data obtained by nuclear sorting is highly reproducible.

## Factors that impact human cell-type specific gene expression

We next investigated the relationship between human cell-type specific gene expression and biological or clinical factors that might influence cellular function. Although the cerebellar samples characterized here were obtained from normal controls, differences in gender, age, and the time between death and tissue preservation (autolysis time or postmortem delay interval) might influence the expression data we obtained. We first analyzed the effect of postmortem delay (PMD) as a potential source of technical variability that can complicate biological interpretations of the data. We found that only a few genes significantly change with PMD: 12 across all samples, four in granule cells, one in basket cells, and one in glia (*Figure 6—figure supplement 1A*, *Supplementary file 8*). Examination of these genes revealed that only one, *KIF19* in glia, changed linearly with PMD (*Figure 6—figure supplement 1C*). The remaining genes appear to increase in only a few samples with intermediate PMDs (*Figure 6—figure supplement 1B*). These data confirm an earlier report that autolysis has a relatively minor effect on expression data obtained from post-mortem human tissue (*Gupta et al., 2012*).

## Gender

To examine the effect of gender on our data, we compared gene expression in samples from males and females and identified genes that are differentially expressed (*Figure 6—figure supplement 1A*, *Supplementary file 8*). As expected, we found that the majority of gender-specific genes resided on the X or Y chromosomes, with expression of X-chromosome genes enriched in female samples and Y-chromosome genes enriched in male samples. Of the 27 genes exhibiting gender-specific differential expression (adjusted p-value<0.01, baseMean >50), seven are located on the X chromosome, 14 are on the Y chromosome, and six on autosomes. Interestingly, gender-specific genes can be more differentially expressed in some cell types than in others. For example, we identified Glycogenin 2 Pseudogene 1 (*GYG2P1*) as a male-enriched gene across all samples, although it is clearly more male-enriched in basket cells than granule cells and glia (*Figure 6A*). This is not due

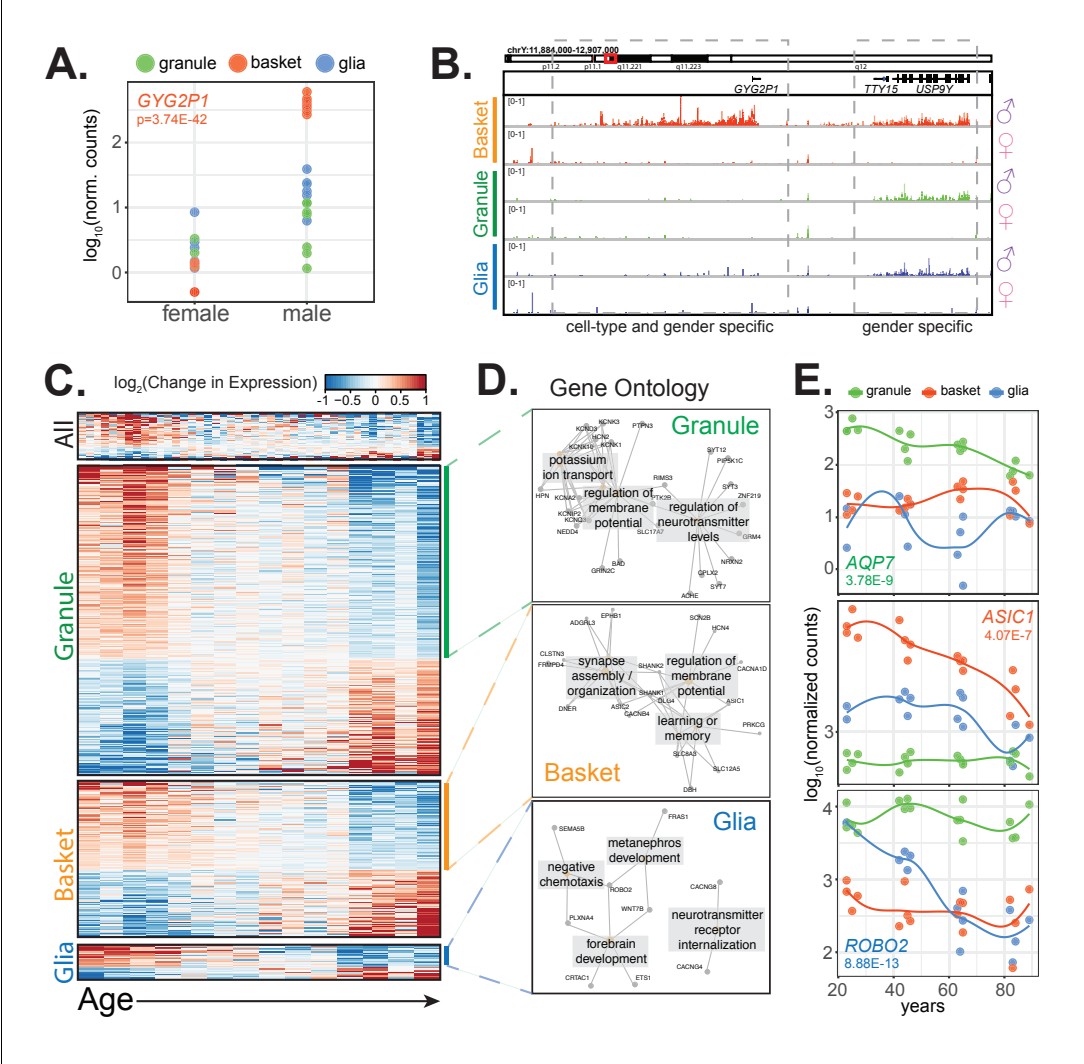

**Figure 6.** Clinical factors impact gene expression in a cell-type specific manner. (A,B) *GYG2P1*: a basket-specific, male-specific gene. (A) quantification of *GYG2P1* gene expression in granule, basket, and glial nuclei for female and male samples. (B) browser view showing expression of male-specific genes *TTY15* and *USP9Y* and the male-specific and basket-specific gene *GYG2P1*. (C–E) Genes that significantly change with age. (C) Heatmap showing change in gene expression across all, granule, basket, or glia nuclei. Columns: samples are sorted by age. Rows: genes ranked by change in gene expression over age. (D) Enrichment map of significant gene ontology categories for aging down-regulated genes from granule cells, basket cells, and glia. (E) Scatterplot showing gene expression across age for three genes that are cell-type specific and down-regulated with age. See also *Figure 6—figure supplement 1*.

DOI: https://doi.org/10.7554/eLife.37551.015

The following figure supplement is available for figure 6:

**Figure supplement 1.** Analysis of clinical factors that contribute to gene expression variability across individuals, related to *Figure 6*.

DOI: https://doi.org/10.7554/eLife.37551.016

to mis-mapped reads as a result of duplication, as Glycogenin 2 (*GYG2*) is expressed only in glia and does not exhibit gender-specific expression (*Figure 6—figure supplement 1D*). Examination of the Y chromosome locus containing *GYG2P1* reveals male-specific basket cell expression across a large region, including *GYG2P1* and approximately 300 kb downstream (*Figure 6B*). In contrast, the nearby genes *TTY15* and *USP9Y* are also male-enriched but they are not cell type specific. Genes that exhibit gender and cell-type specific expression in both glia (*Figure 6—figure supplement 1E*) and granule cells are also evident (*Figure 6—figure supplement 1F*). While the functional significance of these changes is unknown, our results suggest another potential source of sexually dimorphic, cell specific function in the mammalian brain.

## Age

A fundamental question in aging research is whether aging occurs uniformly across the brain, or whether some cell types are more susceptible to the effects of aging than others. To address this issue, we examined the effects of age on gene expression in each of the three cerebellar cell types. We performed differential expression analysis using age as a numerical variable to find genes that significantly change with age in all samples, or specifically in granule cells, basket cells, or glia. We found 274 significant genes in granule cells, 139 in basket cells, 31 in glia, and 42 that are significant across all samples regardless of cell type (*Figure 6—figure supplement 1A*, *Supplementary file 8*). These genes are about equally split between those that increase in expression and those that decrease in expression with age (*Figure 6C*, S9G). Interestingly, only a few genes significantly change with age in more than one cell type. For example, the 42 genes that are identified as significantly changed with age across all cell types appear heterogeneous (*Figure 6C*), and most of these do not reach significance when tested in each cell type individually (*Figure 6—figure supplement 1G*). Finally, while as expected we observe high correlation between age and expression when we examine aging genes in their respective cell type, there is only a modest correlation when we examine these genes in other cell types (*Figure 6—figure supplement 1H*). For example, the Pearson's correlation between age and the expression of the 274 granule aging genes in granule neurons is 0.76, but only 0.42 and 0.32 when expression of these genes is examined in basket neurons and glia. These data suggest that the specific molecular consequences of aging, at least in the cerebellum, are different between cell types.

Despite differences in the genes impacted by aging in each cell type, it remained possible that the cellular processes altered by these changes are related between cell types. To test this hypothesis, we performed GO analysis on the genes that are up- or down-regulated with age in each cell type. We did not find significant GO categories for any group of aging up-regulated genes, nor did we find significant GO categories for the aging down-regulated genes from the all cell types group. However, we found that the genes down-regulated with age in all three individual cell types, despite containing different specific genes, were enriched for synaptic genes (*Figure 6D,E*). Our finding that genes encoding synaptic components decline in expression with age is consistent with previous findings that axons and dendrites atrophy in the brains of older individuals (*Burke and Barnes, 2006*; *Dickstein et al., 2013*; *Freeman et al., 2008*). However, our data extend these observations to establish that this general process impacts distinct genes in each cell type, as might be expected from the distinct protein compositions of synapses or synapse-related functions in each cell type. Although further studies are necessary to determine whether these principles hold for other regions of the brain, our findings argue that a deeper understanding of neurological aging requires examination of defined cell types rather than whole tissue.

## Other factors

To identify inter-individual variability in gene expression that may not be explained by clinical factors, we performed principal component analysis separately for each cell type (*Figure 7A*). We found that while the first principal component for all cell types splits the samples based on gender, the second principle component for granule cells and glia splits three donors (WI, VM, and ZH) away from the other 13 (*Figure 7B*, right panels). Examination of the loading vectors for PC2 from granule and glia samples shows the genes that contribute the most to the separation of these three donors include the immediate early genes *FOS*, *FOSB*, and *NPAS4* as well as heat shock genes *HSPA1A* and *HSPB1* (*Figure 7B*, left panels). Interestingly, while *FOS* and *NPAS4* are important factors for defining PC4 in basket cells, a slightly different group of donors are affected (WI, ZH, TM, and KE) suggesting that variable expression of these genes in different individuals is also cell type specific (*Figure 7—figure supplement 1A*).

To further explore this inter-individual variability, we performed differential expression analysis to identify additional genes that are significantly up- or down-regulated in these three donors (WI, VM, ZH) compared to the others (*Figure 7C*, *Supplementary file 8*). In granule cells, samples WI, VM, and ZH are characterized by the induction of 224 genes and diminished expression of 10 genes. An overlapping but weaker response that included 136 induced genes was present in glial cells. The induction of these genes in glia is not due to contamination by granule cells as highly expressed

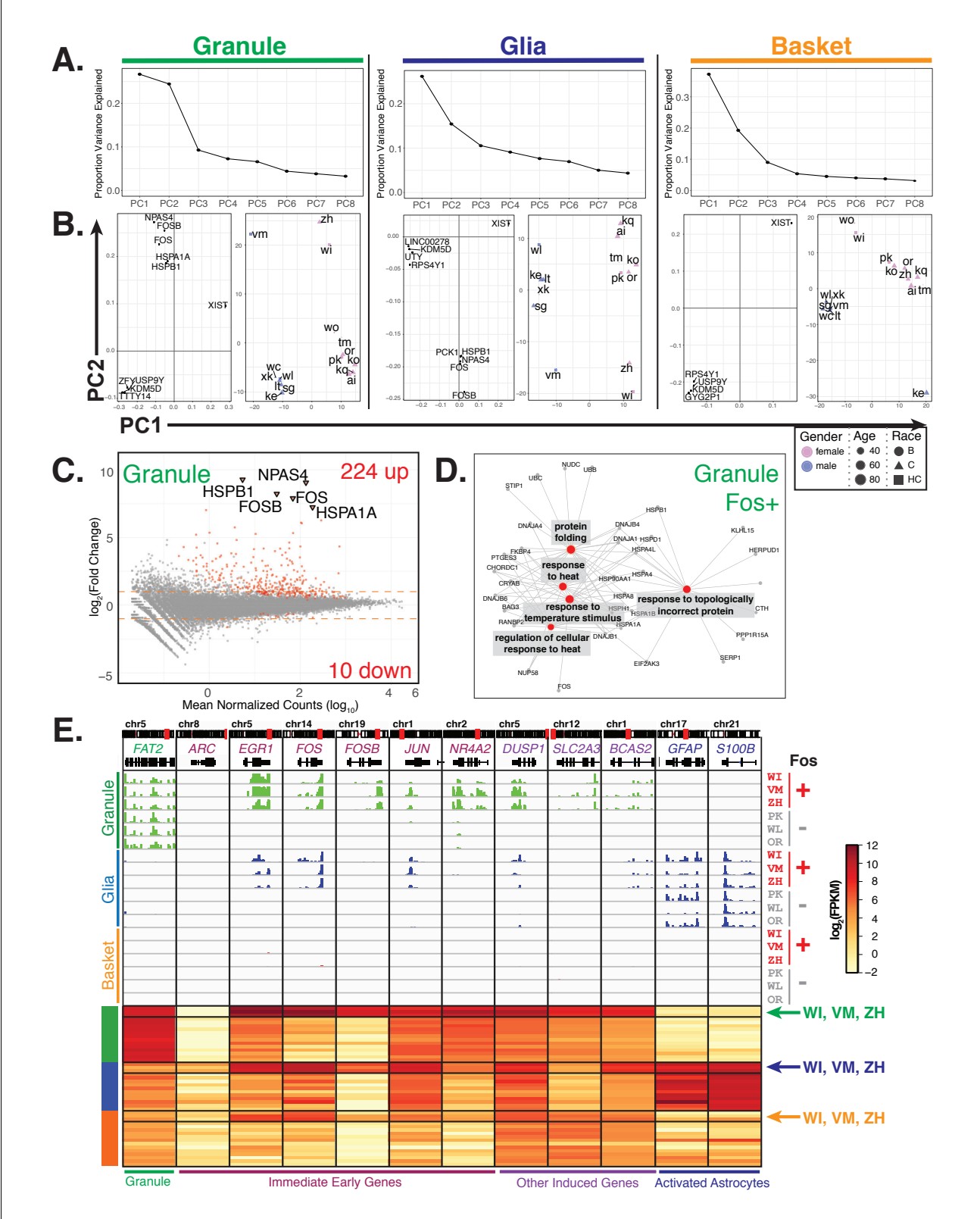

**Figure 7.** Additional sources of interindividual variability in gene expression. (**A, B**) Principal component analysis of samples for each cell type identifies interindividual gene expression variability. (**A**) Plot showing the proportion variance explained by each of the first eight principal components for all granule, glia, and basket cell samples. (**B**) For each cell type, plots showing loadings (left) and scores (right) for PC1 and PC2. For loadings, the five genes with the highest absolute values from the loading vectors for PC1 or PC2 are shown. (**C**) MA-plot showing differential gene expression analysis of

*Figure 7 continued on next page*

*Figure 7 continued*

granule cells between three Fos+ samples versus the other 13 samples. (D) Network diagram of significant enriched GO categories for up-regulated genes in fos+ samples reveals enrichment for genes involved in stress response and protein folding. (E) Browser view showing expression of selected genes in three cell types from Fos+ (WI, VM, ZH) and Fos- (PK, WL, OR) human samples that are age and gender matched. Heatmap shows gene expression of selected genes in all human samples (granule – green, glia – blue, basket – orange). Samples from Fos+ donors WI, VM, and ZH are indicated. See also *Figure 7—figure supplement 1*.

DOI: https://doi.org/10.7554/eLife.37551.017

The following figure supplement is available for figure 7:

**Figure supplement 1.** Additional analyses of interindividual variability in gene expression, related to *Figure 7*.

DOI: https://doi.org/10.7554/eLife.37551.018

markers for granule cells (e.g. *FAT2*) are not present in the glial data (*Figure 7E*). Only two differentially expressed genes were evident in the basket cell data.

To control for the possibility that a random selection of individuals would result in a similar number of differentially expressed genes, we performed differential expression analysis for granule cell samples on all 560 combinations that partition the 16 individuals into two groups containing three and thirteen individuals. Out of the 560 combinations, 529 resulted in fewer than 50 differentially expressed genes, and only five resulted in greater than 234 differentially expressed genes (*Figure 7—figure supplement 1B*). These five combinations all contain groups of three young or old individuals, suggesting that the differentially expressed genes reflect a real biological difference in age-associated genes (*Figure 7—figure supplement 1C*). Furthermore, that the vast majority of combinations result in far fewer than 234 differentially expressed genes suggests that these differences were unlikely to have occurred by chance.

GO analysis of the 224 granule cell induced genes revealed categories associated with protein folding and response to heat (*Figure 7D*, *Supplementary file 9*). Analysis of the expression of immediate early genes, which are associated with neuronal activation (*Okuno, 2011*; *Pérez-Cadahía et al., 2011*), indicated that their induction is also selective. For example, induction of *EGR1*, *FOS*, and *FOSB* occurs, but no difference in expression of *ARC* (a well-characterized immediate early gene) is evident (*Figure 7E*). Furthermore, although expression of immediate early genes such as *c-FOS* can become induced in response to damage such as ischemic stroke, we found no evidence for induction of glial genes such as *GFAP* and *S100B* that mark astrocyte activation in response to stroke and other forms of damage (*Choudhury and Ding, 2016*; *Ding, 2014*; *Dirnagl et al., 1999*; *Kajihara et al., 2001*; *Pekny and Nilsson, 2005*). Finally, examination of the clinical records failed to reveal features (gender, age, PMD, cause of death) that distinguished these three samples from the 13 other control donors.

Although the cause or causes of these molecular events remain obscure, the discovery of this strong and highly reproducible response in select postmortem human brain samples indicates that nuclear expression profiling of specific cell types can be used to discover pathways and biomarkers. Further work is required to determine whether these pathways and biomarkers are associated with human physiological conditions, whether they reflect agonal factors prior to death or technical differences in tissue processing post mortem.

## Discussion

The genes expressed in each cell type define their function and their responses to internal and external cues. Here we report a robust strategy for cell-type specific expression and epigenetic profiling of human postmortem brain that incorporates features of several prior protocols for single cell or population-based studies of CNS cell types. We demonstrate that antibodies against cell specific ER or membrane proteins can often be used to sort isolated nuclei, increasing substantially the candidate antibodies that can be employed for this purpose. We report highly accurate and comprehensive cell-type specific gene expression profiling of neurons and glia from mouse, rat, and human brains. Using this methodology, we find that even classically defined, cerebellar cell types differ between mouse and human by expression of hundreds of orthologous genes. Species specific expression is confirmed by analysis of single nucleus transcriptome data from the same mouse (*Saunders et al., 2018*) and human (*Lake et al., 2018*) cell types, and by ATAC-seq assays

demonstrating that active expression is accompanied by enhanced chromatin accessibility. Interestingly, genes expressed in human but not mouse granule neurons are expressed frequently in other regions of the mouse brain. Analysis of cell-type specific data from sixteen human postmortem brains reveals that the specific molecular consequences of aging differ between cell types, although in each case, expression of genes involved in synapse development and maintenance is diminished. Finally, we report the induction of a robust and cell-type specific molecular response of unknown etiology in granule cells from three of sixteen donors. Taken together, the experiments we present document a precise methodology for comprehensive analysis of specific cell types in rodent and human brains, and demonstrate that this approach can provide an improved avenue for investigation of molecular events associated with human cell type function and dysfunction.

## Species and cell type specific gene expression in the mammalian brain

Although a consensus definition of cell type has yet to emerge for the mammalian CNS, a recent evolutionary model (*Arendt, 2008*; *Arendt et al., 2016*) is helpful for consideration of the data we have generated here. According to this model, the specific characteristics of homologous cell types can vary as long as they remain defined by a distinctive, shared regulatory apparatus. For example, granule cell or astrocyte gene expression profiles can vary between species, or even in individual cells of a type, without losing their cell type identity. This definition can accommodate both functional changes in cell types between species and altered expression of genes within the cell type due to mutations in *cis*-regulatory sequences. Our data document important differences in the expression of orthologous genes in each cell type between rodent and human brains. Hundreds of these differences are cell type specific and, in some cases, these genes remain expressed in other regions of the brain. We believe these data are exciting and provocative. They demonstrate that the fine-tuned biochemistry of homologous human and mouse CNS cell types is likely to differ significantly, and they argue strongly that differences in gene expression profiles cannot be used as the defining criterion for cell type identity between species.

The results presented here contribute three additional insights into mammalian brain evolution. First, while they are consistent with previous published examples of *cis*-regulatory sequence divergence as a mechanism of evolution (*Maricic et al., 2013*; *Prud'homme et al., 2006*; *Weyer and Pääbo, 2016*), the lack of highly enriched GO categories for most of the cell specific events we have documented suggests that regulatory sequence changes are major drivers of phenotypic variation between homologous cell types in mammalian brains. This is consistent with the evolutionary model presented above, since it allows substantial evolution of the functional properties of the cell while maintaining the core regulatory complex (CoRC) that defines a given cell type (*Arendt et al., 2016*). Second, the expression differences we have documented are large, and we have restricted our analysis to high confidence orthologous genes. Our data, therefore, complement a previous study of progenitor cells that reported substantive changes in the expression of human genes that lack mouse orthologues (*Florio et al., 2015*). We find also that changes in gene expression between species are accompanied by changes in promoter chromatin accessibility, and that DNA sequences at putative regulatory regions around species-enriched genes are less conserved that for unchanged genes. While our work is consistent with previous studies showing that the evolution of promoters is slower than for other regulatory elements (*Bae et al., 2015*; *De la Torre-Ubieta et al., 2018*; *Villar et al., 2015*), our finding suggest that promoter evolution also plays an important role in regulation of cell-type specific gene expression. We note that in mouse enriched granule cell expressed genes, the preferential occurrence of GATA motifs in the gene body and mouse specific expression of the atypical GATA binding factor *Trps1* suggest also that differential transcription factor expression and regulatory domains within genes may contribute to divergent transcription between species. Taken together, these datasets predict that the expressed differences between mouse and human cell types are extensive, cell-type specific, and phenotypically important.

## Molecular phenotyping of human CNS cell types

Recent advances in technologies for human genome sequencing and analysis have led to an astounding increase in our knowledge of the complex genetic causes of human psychiatric and neurological disease (*Burguière et al., 2015*; *Hinz and Geschwind, 2017*; *Vorstman et al., 2017*). In some cases, recreation of the causative mutations in the mouse genome has resulted in experimental

models that are sufficiently accurate for investigation of molecular basis of the disorder (*Lombardi et al., 2015*; *Lyst and Bird, 2015*; *Orr, 2012*), and they have led to the discovery of unexpected features of the disease (*Baker et al., 2013*; *Guy et al., 2007*). In other cases, informative animal models remain elusive (*Lavin, 2013*), and investigation of molecular mechanisms of disease difficult (*Biton et al., 2008*; *Medina and Avila, 2014*). Here we present data for three cerebellar cell types in sixteen human brain samples. All samples were obtained from unaffected subjects that received a neuropathological diagnosis of 'normal adult brain'. The correlation coefficients between samples (r = 0.98–1.0) indicated that the data are extremely reproducible, and that the impact of PMD on the expression profile of nuclear RNAs is minor. Initial indications that gender and age impact gene expression differently in CNS cell types are interesting, and they suggest that nuclear profiling is sufficiently accurate for detailed investigation of both sexually dimorphic human behaviors and the aging of specific brain circuitry. An exciting possibility, for example, is that comparative analysis of human brains may provide critical insight into the impacts of aging on cell types that are selectively vulnerable into late-onset human disorders.

One of the most intriguing findings we present is that cerebellar granule cells from three brain samples exhibit robust induction of a set of 224 genes containing many immediate early genes, and that are the set is enriched for GO categories (protein folding/refolding, apoptosis, transcriptional response to stress, response to external stimuli, ATPase activity, etc.) that indicate a shared, acute response to an unknown physiological stimulus or an unexpected and robust event occurring during sample processing. As stated above, no correlation with PMD and no induction of glial markers indicative of stroke or brain damage is evident in our data. Although activity dependent gene expression changes have been characterized extensively in cultured mouse granule cells, the response identified in these three human samples is overlapping but distinctive. While we do not wish to speculate on the cause of this response, our data demonstrate that careful comparative studies of cell specific datasets from control samples is an important feature of experimental design for comparative analysis of molecular events associated with responses to adverse genetic or environmental events.

Finally, our data provide additional examples of the importance of epigenetic mapping in specific cell types as a complement to gene expression analysis. In particular, the use of ATAC-seq mapping (*Buenrostro et al., 2015*; *Chen et al., 2016*) of putative regulatory domains for specific CNS cell types confirms that actively expressed genes contain accessible chromatin domains. The lack of these domains in a gene that is not transcribed in the homologous cell type from another species demonstrates that lack of expression reflects complex mechanisms that establish a repressed domain of chromatin proximal to the gene. Although our data do not identify the specific regulatory domains associated with these changes in expression, precedent from studies of enhancer evolution (*Villar et al., 2015*) suggest that further investigation of differentially accessible in these genes can lead to a mechanistic understanding of species specific gene expression.

## Concluding remarks

The driving force for development of an accurate and efficient non-genetic method for characterization of the molecular properties of defined CNS cell types is to enable direct investigation of human biology. Given the tremendous explosion of studies identifying the genetic causes of human disease and the demonstrations in both animal models and humans that these can include lesions that involve simple changes in gene dosage, exploration of details of cell types in normal and affected brains is imperative. We believe comparative analysis of human cell type specific data with that obtained in experimental systems is an important approach toward advancing our understanding of the large variety of human disorders that impact CNS circuits.

## Materials and methods

**Key resources table**

| Reagent type (species) or resource | Designation | Source or reference | Identifiers | Additional information |
|---|---|---|---|---|
| Strain (*M. musculus*) | NeuroD1 EGFP-L10a | (*Doyle et al., 2008*) | RRID:IMSR_JAX:030262 | |

*Continued on next page*

*Continued*

| Reagent type (species) or resource | Designation | Source or reference | Identifiers | Additional information |
|---|---|---|---|---|
| Strain (*M. musculus*) | Pcp2 EGFP-L10a | (*Doyle et al., 2008*) | RRID:IMSR_JAX:030267 | |
| Strain (*M. musculus*) | Sept4 EGFP-L10a | (*Doyle et al., 2008*) | RRID:IMSR_JAX:030271 | |
| Strain (*M. musculus*) | Glt25d2 EGFP-L10a | (*Doyle et al., 2008*) | RRID:IMSR_JAX:030257 | |
| Strain (*M. musculus*) | Ntsr1 EGFP-L10a | (*Doyle et al., 2008*) | RRID:IMSR_JAX:030264 | |
| Strain (*M. musculus*) | Wild type | Jackson labs | RRID:IMSR_JAX:000664 | |
| Antibody | anti-NeuN | Abcam | RRID:AB_2532109 | Rabbit monoclonal; 1:250 for nuclei staining; 1:250 for IF |
| Antibody | anti-NeuN | Millipore | RRID:AB_2298772 | Mouse monoclonal; 1:250 for nuclei staining; 1:250 for IF |
| Antibody | anti-Itpr1 | Abcam | ab190239 | Mouse monoclonal; 1:500 for nuclei staining; 1:500 for IF |
| Antibody | anti-Sorcs3 | Thermo | RRID:AB_2606387 | Goat polyclonal; 1:250 for nuclei staining; 1:250 for IF |
| Antibody | anti-EAAT1 | Abcam | RRID:AB_304334 | Rabbit polyclonal; 1:250 for nuclei staining |
| Antibody | anti-Olig2 | R and D | RRID:AB_2157554 | Goat polyclonal; 1:250 for nuclei staining |
| Antibody | anti-GFAP | Abcam | RRID:AB_880202 | Goat polyclonal; 1:500 for IF |
| Antibody | anti-Mog | Thermo | RRID:AB_2607363 | Goat polyclonal; 1:500 for IF |
| Antibody | anti-Mag | Cell Signaling | RRID:AB_2665480 | Rabbit polyclonal; 1:500 for IF |
| Antibody | anti-Pde1a | Acris | TA311317 | Goat polyclonal; 1:200 for IF |
| Antibody | anti-Pde1c | Santa Cruz | RRID:AB_11149544 | Rabbit polyclonal; 1:100 for IF |
| Sequence-based reagent | raw and processed sequencing data | this paper | GSE101918 | Details about all samples in this superseries are in *Supplementary file 1* |
| Sequence-based reagent | raw and processed sequencing data | (*Mo et al., 2015*) | GSE63137 | Details about all samples in this superseries are in *Supplementary file 1* |
| Sequence-based reagent | raw and processed sequencing data | (*Habib et al., 2016*) | GSE85721 | Details about all samples in this superseries are in *Supplementary file 1* |
| Sequence-based reagent | raw and processed sequencing data | (*Lake et al., 2018*) | GSE97930 | Details about all samples in this superseries are in *Supplementary file 1* |
| Sequence-based reagent | raw and processed sequencing data | (*Saunders et al., 2018*) | GSE116470/dropvis.org | Details about all samples in this superseries are in *Supplementary file 1* |
| Commercial assay or kit | AllPrep FFPE | Qiagen | 80234 | |
| Commercial assay or kit | Rneasy Micro | Qiagen | 74004 | |
| Commercial assay or kit | MinElute Reaction Cleanup | Qiagen | 28206 | |
| Commercial assay or kit | Ovation RNAseq V2 | Nugen | 7102–32 | |
| Commercial assay or kit | Ultra II DNA Lirary Prep Kit for Illumina | NEB | E7645L | |
| Commercial assay or kit | Multiplex Adapters for Illumina | NEB | E7335L, E7500L | |
| Commercial assay or kit | Bioanalyzer Pico chips | Agilent | 5067–1513 | |
| Commercial assay or kit | TapeStation D1000 ScreenTape | Agilent | 5067–5583 | |

*Continued on next page*

*Continued*

| Reagent type (species) or resource | Designation | Source or reference | Identifiers | Additional information |
|---|---|---|---|---|
| Commercial assay or kit | TapeStation High Sensitivity D1000 ScreenTape | Agilent | 5067–5585 | |
| Software, algorithm | source code | (*Xu, 2018*) | https://github.com/ xu-xiao/ non_transgenic_cell_ type_profiling | R scripts for analysis and generating figures; parameters for running command line tools. |

## Animals

All animal protocols were carried out in accordance with the US National Institutes of Health Guide for the Care and Use of Laboratory Animals and were approved by the Rockefeller University Institutional Animal Care and Use Committee. All mice were raised at 78 °F in 12 hr light:12 hr dark conditions with food and water provided ad libitum. Rat tissues were provided by Winrich Freiwald. Strain information is detailed in *Supplementary file 1*.

## Summarized methods

Below are summarized procedures for nuclei isolation and labeling, RNA-seq and ATAC-seq library preparation. A full protocol detailing all steps can be found in the section Detailed protocol for nuclei isolation, staining, RNA-seq and ATAC-seq.

## Nuclei isolation

Nuclei isolation was adapted from the protocol described in previous publications (*Kriaucionis and Heintz, 2009*; *Mellén et al., 2012*). For mouse experiments, cortex and cerebella were dissected as described previously. All tissue from C57BL/6J animals were flash frozen using liquid nitrogen. Tissue from bacTRAP animals were freshly dissected and directly used for homogenization. Rat and human tissue were obtained frozen. All frozen tissues were thawed on ice for a minimum of 30 min before use.

To isolate nuclei, fresh or thawed tissue were transferred to 5 mL of homogenization medium (0.25 M sucrose, 150 mM KCl, 5 mM MgCl2, 20 mM Tricine pH 7.8, 0.15 mM spermine, 0.5 mM spermidine, EDTA-free protease inhibitor cocktail, 1 mM DTT, 20 U/mL Superase-In RNase inhibitor, 40 U/mL RNasin ribonuclease inhibitor). Tissue were homogenized by 30 strokes of loose (A) followed by 30 strokes of tight (B) glass dounce. Homogenate was supplemented with 4.6 mL of a 50% iodixanol solution (50% Iodixanol/Optiprep, 150 mM KCl, 5 mM MgCl2, 20 mM Tricine pH 7.8, 0.15 mM spermine, 0.5 mM spermidine, EDTA-free protease inhibitor cocktail, 1 mM DTT, 20 U/mL Superase-In RNase inhibitor, 40 U/mL RNasin ribonuclease inhibitor), and laid on a 27% iodixanol cushion. Nuclei were pelleted by centrifugation 30 min, 10,000 rpm, 4°C in swinging bucket rotor (SW41) in a Beckman Coulter XL-70 ultracentrifuge. For mouse, rat, and human cytoplasmic fractions, 175 uL of the top layer was added to 175 uL Qiagen Buffer RLT and stored at −80°C. The nuclear pellet was resuspended in homogenization buffer.

## Nuclei labeling and sorting

After nuclei isolation, resuspended nuclei were fixed with 1% formaldehyde for 8 min at room temperature, and then quenched with 0.125M glycine for 5 min. Nuclei were pelleted at 1000 g, 4 min, 4°C, and then washed two times with Wash Buffer (PBS, 0.05% TritonX-100, 50 ng/mL BSA, 1 mM DTT, 10 U/uL Superase-In RNase Inhibitor). Nuclei were blocked with Block Buffer (Wash buffer with an additional 50 ng/mL BSA) for 30 min, incubated with primary antibody for 1 hr, and then washed three times with Wash Buffer with spins in between washes as described above. Nuclei were then incubated in secondary antibody for 30 min and washed three times with Wash Buffer. All incubations steps were performed at room temperature. Primary and secondary antibodies were diluted in Block Buffer. Secondary antibodies were purchased from Life Technologies or Jackson Immunoresearch and were used at 1:500 dilution. Secondary antibodies from goat used: mouse Alexa488, rabbit Alexa594. Secondary antibodies from donkey used: mouse Alexa488, goat Alexa488, rabbit Alexa488, mouse Alexa594, rabbit Alexa594.

## Primary antibodies

Antibody concentrations for flow cytometry were determined empirically for mouse cerebellar nuclei by finding the lowest concentration that reproducibly separated populations of interest from other nuclei. We kept these same concentrations for rat and human nuclei. While we do observe differences in cross-species reactivity of antibodies, in general, we found that these differences made it easier to identify and gate for distinct populations in rat and human compared to mouse (e.g. both NeuN antibodies result in clear positive and negative populations in rat and human, but this difference is less clear in mouse). To ensure that differences in antibody reactivity does not affect the purity of our sorted populations, we used post-hoc analysis to rigorously validate that each dataset contains markers for the target cell type and do not contain markers for other cell types (*Figure 1C–D*, *Figure 2B*, *Figure 1—figure supplement 2C*, *Figure 5B*).

| Antigen | Species | Vendor | Cat. # | RRID | Dilution |
|---|---|---|---|---|---|
| NeuN | Rabbit | Abcam | ab177487 | RRID:AB_2532109 | 1:250 |
| NeuN | Mouse | Millipore | MAB377 | RRID:AB_2298772 | 1:250 |
| Itpr1 | Mouse | Abcam | ab190239 | | 1:500 |
| Sorcs3 | Goat | Thermo | PA5-48023 | RRID:AB_2606387 | 1:250 |
| EAAT1 | Rabbit | Abcam | ab416 | RRID:AB_304334 | 1:250 |
| Olig2 | Goat | R and D | AF2418 | RRID:AB_2157554 | 1:250 |

Antibody combinations used for isolating all cell types are below. Unless indicated, NeuN refers to the rabbit antibody.

## Mouse

Granule: Itpr1, Sorcs3, Olig2, NeuN. Purkinje: Itpr1, NeuN. Basket: Sorcs3, NeuN. Astrocyte: EAAT1, NeuN (mouse). All oligodendroctyes, mature oligodendrocyte, OPC: Olig2, NeuN.

## Rat

Granule: Itpr1, Sorcs3, Olig2. Purkinje and basket: Itpr1, NeuN. Astrocyte and oligodendrocytes: Sorcs3, NeuN. OPC: Olig2, NeuN.

## Human

Granule, basket, and glia: Itpr1, NeuN. Astrocyte: Itpr1, Olig2, NeuN. Mature oligodendrocyte, OPC: Olig2, NeuN.

## Flow cytometry

Prior to flow cytometry, nuclei were co-stained with DyeCycle Ruby to 20 uM final concentration. Nuclei were analyzed using a BD LSRII (BD Biosciences, San Jose, CA, USA) flow cytometer using the 488 nm, 561 nm, and 640 nm lasers. Nuclei were sorted using a BD FACSAria cell sorter using the 488 nm, 561 nm, and 635/640 nm lasers. All samples were first gated using DyeCycle Ruby to determine singlets. Analysis was performed using FACSDiva (BD) or FlowJo software. Qiagen Buffer RLT (unfixed samples) or Qiagen Buffer PKD (fixed samples) were added to the sorted nuclei and the samples were stored at −80℃. Information about percentages for all gated populations can be found in *Supplementary file 1*.

## RNA purification, library construction, and sequencing

RNA from cytoplasmic fractions or fresh nuclei were purified using the Qiagen RNeasy Micro kit with on-column DNase digestion. RNA from fixed nuclei were purified using the Qiagen RNeasy FFPE kit with the following modifications – after Proteinase K digestion, RNA was spun at max speed for 15 min. The supernatant was removed and incubated at 65℃ for 30 min, 70℃ for 30 min, and 80℃ for 15 min. An on-column DNase digestion was performed in place of the in-solution DNAase digestion. RNA quantity was determined using the Qubit RNA HS Assay kit and RNA quality was determined using Agilent 2100 Bioanalyzer with RNA 6000 Pico chips. Purified RNA was converted to cDNA and

amplified using the Nugen Ovation RNA-seq System V2. cDNA was fragmented to an average size of 250 bp using a Covaris C2 sonicator with the following parameters: intensity 5, duty cycle 10%, cycles per burst 200, treatment time 120 s. Libraries were prepared using the Illumina TruSeq DNA LT Library Prepartion Kit or the NEBNext Ultra DNA Library Prep Kit for Illumina with NEBNext Multiplex Oligos for Illumina. The quality of the libraries was assessed using the Agilent 2200 TapeStation system with D1000 High Sensitivity ScreenTapes. Libraries were sequenced at The Rockefeller University Genomics Resource Center on the Illumina HiSeq 2500 machine to obtain 50 bp single-end reads or on the Illumina NextSeq 500 to obtain 75 bp paired-end reads. A summary of all RNA-seq datasets can be found in *Supplementary file 1*. All datasets have been deposited in GEO: GSE101918.

## ATAC-seq library preparation and sequencing

ATAC-seq libraries from formaldehyde fixed nuclei were prepared essentially as described (*Buenrostro et al., 2013*; *Buenrostro et al., 2015*; *Chen et al., 2016*), with minor modifications. 50,000 sorted nuclei from human granule, human basket, mouse granule, or mouse basket nuclei were isolated as described above and incubated for 10 min in lysis buffer (10 mM Tris pH 7.5, 10 mM NaCl, 3 mM MgCl2, 0.1% NP-40). Nuclei were then resuspended in 50 ul 1x TD Buffer containing 2.5 ul Tn5 enzyme from Nextera kit (Illumina). Transposition reactions were incubated for 30 min at 37°C and mixed with 200 uL of a reverse-crosslinking buffer (50 mM Tris–Cl, 1 mM EDTA, 1% SDS, 0.2 M NaCl, 5 ng/ml proteinase K). The reaction was incubated overnight at 65°C with 1000 r. p.m. shaking, and then purified with QiaQuick MinElute columns (Qiagen). Purified DNA was amplified by PCR using Q5 High Fidelity Polymerase (NEB) for 12 cycles with barcoded primers, as described (*Buenrostro et al., 2013*). Amplified libraries were purified and size selected with AMPure XP beads (Beckman Coulter). Libraries were sequenced on Illumina HiSeq 2500 to yield 50 bp paired-end reads and on Illumina NextSeq 500 to yield 75 bp paired-end reads. Raw data was uploaded to Illumina basespace to separate reads based on barcoded sequences and generate individual FastQ files for each sample.

## Detailed protocol for nuclei isolation, staining, RNA-seq andATAC-seq

This protocol is a general guide for staining nuclei for flow cytometry (analysis or sorting). The nuclei isolation protocol (I) is adapted from *Kriaucionis and Heintz (2009)* with slight changes to the percentages of iodixanol in the density gradient, spin speed, and spin times. Nuclei fixation and staining (II) is based generally on protocols for intracellular staining of cells for flow cytometry. Flow cytometry analysis/sorting (III) is performed as usual except that the nucleic acid labeling fluorophore Dye-Cycle Ruby is added to distinguish single nuclei from aggregates. Preparation of libraries for ATAC-seq (IV) is adapted from *Chen et al. (2016)* and *Buenrostro et al., 2015*. RNA isolation (**V**) uses Qiagen FFPE kits with modifications. cDNA production (VI) uses Nugen's Ovation RNAseq V2 kit. Library construction (VII) involves sonication of cDNA then end repair, a-tailing, adapter ligation, and fragment enrichment. The protocol here will describe how to do this using the NEB Ultra II DNA Library Prep Kit for Illumina with NEBNext Multiplex Oligos for Illumina, although I have also had success using Illumina's TrueSeq kits as well as off-the-shelf enzymes from NEB along with oligos from IDT.

Modifications should be made depending on whether sorted nuclei are to be used for analysis or RNA isolation (use buffers for RNA). Stained nuclei can be stored on ice at 4°C (cover with foil) for 1–2 days before sorting.

### Buffers

(store Buffers A and B at RT; make all other buffers fresh; Buffers C and D can be made the night before, but always add spermidine, spermine, protease inhibitors, DTT, and RNase inhibitors fresh; use RNase free water if isolating RNA, otherwise can use autoclaved MilliQ water)

### Buffer A
Stock Optiprep 60% iodixanol (light sensitive; keep wrapped in foil in covered location)

| Buffer B | Stock | **50 mL** |
|---|---|---|
| 900 mM KCl | 2M | 22.5 mL |
| 30 mM MgCl$_2$ | 1M | 1.5 mL |
| 120 mM Tricine-KOH, pH 7.8 | 0.5M | 12 mL |
| Water | | 14 mL |
| **Buffer C** | Stock | **40 mL** |
| 5 volumes Buffer A | | 33.33 mL |
| 1 vol Buffer B | | 6.67 mL |
| 0.5 mM spermidine | 500 mM | 40 uL |
| 0.15 mM spermine | 100 mM | 60 uL |
| Protease inhibitors (EDTA free) | | four pills |
| 1 mM DTT | 1M | 40 uL |
| Superasin | | 40 uL |
| **Buffer D** | Stock | **50 mL** |
| 0.25M sucrose | 2.5M | 5 mL |
| 150 mM KCl | 2M | 3.75 mL |
| 5 mM MgCl$_2$ | 1M | 500 uL |
| 20 mM Tricine-KOH, pH 7.8 | 0.5M | 2 mL |
| Water | | up to 50 mL |
| 0.5 mM spermidine | 500 mM | 50 uL |
| 0.15 mM spermine | 100 mM | 75 uL |
| Protease inhibitors (EDTA free) | | five pills |
| 1 mM DTT | 1M | 50 uL |
| Superasin | | 50 uL |
| RNasin | | 50 uL |
| **Buffer E: 27% iodixanol** | | **13** mL |
| Buffer C | | 7.02 mL |
| Buffer D | | 5.98 mL |
| **Wash/Block buffer for analysis** - 3% BSA in PBS with 0.05% TritonX-100 | | |
| **Wash buffer for RNA** | Stock | **50 mL** |
| PBS | | 50 mL |
| 50 ng/mL BSA | 50 ug/mL | 50 uL |
| 0.05% TritonX-100 | 10% | 250 uL |
| 1 mM DTT | 1M | 50 uL |
| Superasin | | 25 uL |
| **Block buffer for RNA** | Stock | **50 mL** |
| Wash buffer for RNA | | 50 mL |
| 100 ng/mL BSA | 50 ug/mL | 50 uL |
| **Cushion** | Stock | |
| Buffer D | | 500 uL |
| Glycerol | 50% | 250 uL |
| NP-40 | 10% | 5 uL |
| **Buffers for ATAC** - from *Chen et al. (2016)* and *Buenrostro et al., 2015* | | |
| **Lysis buffer** | Stock | **1 mL** |
| 10 mM Tris-Cl, pH 7.4 | 1M | 10 uL |

| | | |
|---|---|---|
| 10 mM NaCl | 5M | 2 uL |
| 3 mM MgCl$_2$ | 1M | 3 uL |
| 3 mM MgCl$_2$ | 1M | 3 uL |
| 0.05% NP-40 | 10% | 5 uL |
| Water | | 980 uL |
| **Reverse-crosslinking buffer** | **Stock** | **1 mL** |
| 50 mM Tris-Cl, pH 7.4 | 1M | 50 uL |
| 1 mM EDTA | 0.5M | 2 uL |
| 0.2M NaCl | 5M | 40 uL |
| 1% SDS | 20% | 5 uL |
| 5 ng/mL proteinase K | 1 ug/mL | 5 uL |
| Water | | 898 uL |

## Procedures

I.  Nuclei isolation
    1.  Dissect tissue of interest and place into 4.5 mL of Buffer D, or if using snap frozen tissue, thaw on ice, minimum of 30 min. If tissue is big, after thawing, transfer to dish of cold PBS and dissect as needed. The ratio of tissue to buffer will need to be determined empirically based on size of tissue and number of nuclei expected, but for mice, I generally use 5 mL of buffer for 2 cerebella or two cortices. For human cerebella, 5 mL of buffer for 50–100 mg of tissue.
    2.  In meantime, make and place Buffers C and D on ice. Place dounce homogenizer with loose-fitting pestle (A) inside on ice.
    3.  If using freshly dissected tissue, pour buffer with tissue into homogenizer. If using frozen tissue, pipette 4.5 mL of Buffer D into homogenizer, then transfer tissue using a pair of clean forceps.
    4.  Homogenize tissue using 30 strokes with pestle A followed by 30 strokes of pestle B.
    5.  Pour homogenate into a 15 mL Falcon tube and adjust to 5 mL total volume with Buffer D. Bring to 24% iodixanol by adding 4.6 mL Buffer C to 5 mL homogenate. Mix by inverting 10 times.
    6.  Prepare Buffer E and mix by inverting 10 times. Add 2 mL to the bottom of a 13.2 ultra-clear ultracentrifuge tube. Place tube on ice.
    7.  Slowly layer homogenate on top of Buffer E. <u>This step is important so do it slowly!</u> Place ultraclear tube into Beckman ultracentrifuge bucket.
    8.  Balance the buckets and cap. Difference between tubes must be less than 0.05 g. Use Buffer D to adjust the weights as needed.
    9.  Spin with SW41 rotor, 10,000 rpm, 45 min, 4°C, slowest deceleration.
    10. Use forceps to retrieve samples from buckets and place on ice. Examine the tube for the presence of a pellet at the bottom. If the pellet is sizable, pour out supernatant and resuspend pellet with 500 uL of Buffer D. If pellet is not observed, the nuclei are likely at the interphase. Carefully remove as much supernatant as possible, and then remove the layer at the interphase. Can mount on slide with a little Prolong DAPI mounting media to check for the presence of nuclei.
    11. Supernatant (remove lipid layer that has formed on top first) can be saved for cytoplasmic RNA isolation or for protein for western blots. For RNA, combine 175 uL supernatant with 175 uL Buffer RLT from Qiagen RNeasy kits. Vortex 30 s, let sit at RT 10 min, vortex 30 s, store at −80°C. For protein, quantify protein using Qubit, then adjust to 20% glycerol, mix, aliquot, and store at −80°C.
    12. Nuclei should be resuspended using a plastic transfer pipette then passed through a cell strainer cap.
II. Nuclei fixation and staining
    1.  Fix nuclei by added 16% formaldehyde (no methanol) to 1% final, then incubate at RT for 8 min, gentle rotation, and covered with foil.
    2.  Quench formaldehyde by adding same volume of 2M glycine as added formaldehyde, for a final concentration of 125 mM glycine. Incubate at RT for 5 min, gentle rotation.

3. Spin 4 min, 1000 g, 4°C to pellet nuclei. Pour out supernatant then add 1 mL of Buffer D. Resuspend.
4. Spin 4 min, 1000 g, 4°C, and then pour out supernatant. Add 1 mL of Wash Buffer and <u>transfer</u> nuclei to a new tube.
5. Spin 4 min, 1000 g, 4°C, and then pour out supernatant. Add 500 uL of Block Buffer. Block at RT for 30 min, gentle rotation.
6. Directly add antibodies to nuclei in Block Buffer. Incubate RT for 1 hr, gentle rotation (can also perform this step at 4°C O/N, but might need to be tested for different antibodies).
7. Take a small aliquot of nuclei at this step as unstained control for flow cytometry.
8. Add 1 mL of Wash Buffer to nuclei, mix, then spin 3 min, 1000 g, 4°C. Pour out supernatant and wash twice more with 1 mL of wash buffer.
9. After last wash, add 500 uL of Block Buffer with 1:500 dilution of appropriate secondary antibodies. Incubate RT for 30 min, gentle rotation, covered with foil.
10. Add 1 mL of Wash Buffer to nuclei, mix, then spin 3 min, 1000 g, 4°C. Pour out supernatant and wash twice more with 1 mL of wash buffer.
11. Pour out supernatant. If sorting, resuspend in appropriate amount of Block Buffer. If performing analysis, can usually just resuspend in residual supernatant.
12. Proceed to flow cytometry analysis or store nuclei on ice with tube at an angle. Place ice bucket at 4°C and cover with foil.

III. Flow cytometry analysis/sorting
1. Before performing analysis or sorting, add appropriate amount of DyeCycle Ruby to samples. The amount is based on the number of nuclei but also the volume of buffer and whether the nuclei are from mouse or human. In general, I have used 1–2 uL per mouse cerebella, and 5–7 uL for 200 mg of human tissue. The actual concentration of dcRuby is not critical as the voltage is easily adjusted on the flow cytometer. The rule of thumb is that after mixing the ruby into the nuclei, the suspended nuclei solution should take on a pale purple appearance.
2. For sorting, keep samples covered and on ice. For analysis, it is okay to have samples at RT, and some exposure to dim light is okay (cover if lights are very bright).
3. General guidelines for flow cytometry –
   - Plot FSC against SSC and gate on the general scatter. Adjust voltage of FSC and SSC as needed. Human nuclei are much bigger than mouse nuclei.
   - Plot dcRuby (linear) and adjust voltage so that singlets center around 50, doublets around 100, triplets around 150, etc. Gate on the singlets.
   - For single color (e.g. A488), plot 488C (fluorescence) against 488B (autofluorescence) and gate on non-autofluorescent positives.
   - For double color (e.g. A488 +A594), plot against each other and gate populations as appropriate.
   - Ideally, a clean sample will have over 85% of population in scatter, and over 75% of scatter in singlets. If numbers are lower than this, will lose a lot of nuclei.
   - Always check the percentage of the gated population to see if it makes sense based on expected percentage of cell type of interest.
   - <u>Always</u> perform a post-sort analysis. If gated population is rare, gate on negatives, sort 1 tube of negatives, and perform post-sort on negatives. This step is important for confirming that sorting is working as expected.
   - If nuclei population is rare (<0.5% of total), add a small amount of Cushion to the bottom of collection tube so that the initial sorted nuclei do not dry out.
4. After sort, to store nuclei for RNA isolation, add Buffer PKD (from Qiagen FFPE RNA kits) to 150 uL total volume if nuclei are 100 uL or less. If more than 100 uL, add 50% vol in PKD. Store at −80°C.
5. For ATAC, transfer 50,000 nuclei to a new tube. For samples sorted with a 70 um nozzle, this is around 50 uL sorted nuclei.

IV. ATAC-seq
1. This protocol is adapted *Chen et al. (2016)* and *Buenrostro et al., 2015*. The beginning has been modified to accommodate sorted nuclei rather than cells. For the last cleanup step, XP beads are used in place of MinElute columns so that size selection be be performed.
2. Starting with 50,000 sorted nuclei, centrifuge 5 min at 500 g, 4°C. Remove and discard supernatant.

3. Gently pipet up and down to resuspend the cell pellet in 50 uL of cold lysis buffer. Centrifuge immediately for 10 min at 500 g, 4°C. Discard supernatent and place pellet on ice.
4. Resuspend the pellet in 50 uL of transposition reaction mix (25 uL Nextera reaction buffer, 2.5 uL Nextera TDE1, 22.5 uL nuclease-free water).
5. Incubate the transposition reaction at 37°C for 30 min with gentle shaking.
6. Following transposition, add 200 uL of reverse-crosslinking solution. Incubate at 65°C, 1000 rpm shaking, overnight.
7. Purify with Qiagen MinElute kit and elute in 20 uL EB buffer. Purified DNA can be stored at −20°C. As control, run 2 uL on TapeStation D1000 High Sensitivity ScreenTape to check for proper tagmentation.
8. To amplify transposed DNA fragments, combine in a PCR tube, 10 uL transposed DNA, 10 uL nuclease-free water, 2.5 uL 25 uM PCR Primer 1, 2.5 uL 25 uM Barcoded PCR Primer 2, 25 uL NEBNext High-Fidelity 2x PCR Master Mix. Primers as described by *Buenrostro et al., 2015* were ordered from IDT. PCR conditions: 72°C - 5 min, 98°C – 30 s, (98°C – 10 s, 63°C – 30 s, 72°C - 1 min) x12 cycles, 65°C – 5 min, hold at 4°C.
9. Cleanup with XP beads. Add 45 uL (0.9x) XP beads and mix by pipetting.
10. Incubate RT 5 min, then place on magnetic stand and allow to clump for 5 min.
11. Add 200 uL of 80% EtOH and let stand 30 s. Discard supernatant.
12. Repeat.
13. Remove residual EtOH and air dry 5–15 min.
14. Add 21 uL of Illumina Resuspension Buffer and resuspend beads by pipetting. Incubate at RT for 2 min, then place on magnetic stand and let clump 2 min.
15. Remove 30 uL and transfer to clean tube.
16. Dilute library 1:5 then check size and concentration on TapeStation using High Sensitivity D1000 ScreenTape. Pool libraries.
17. Run another High Sensitivity D1000 ScreenTape with pooled samples to get average peak size and to check that no major pipetting errors occurred during pooling (concentration of pool should be around 10 nM).
18. Submit for sequencing with TapeStation results.

V. RNA (and DNA) isolation
1. Set one heat block to 56°C and another to 65°C. For all heat block steps in this protocol, the exact temperature is important so check using a thermometer that temperature is correct (no more than 1–2 degrees off). Time is also critical here, so make sure to set a timer and DO NOT incubate for longer than indicated or your RNA will be unusable.
2. Thaw nuclei in PKD at RT; briefly spin samples to collect liquid at bottom and add 10 uL of Proteinase K (included in the kit).
3. Mix by inverting a few times then briefly spin. Place in 56°C heat block for 15 min.
4. Transfer samples directly from heat block onto ice and incubate on ice for 3 min.
5. Spin at 20,000 g for 15 min, RT.
6. Transfer supernatant to a new tube, taking care not to transfer the pellet. Place tubes in 65°C heat block. Incubate 30 min.
7. Transfer tubes to 70°C heat block and incubate samples for an additional 30 min.
8. Transfer tubes to 80°C heat block and incubate samples for an additional 15 min.
9. Briefly spin down samples and then allow to cool at RT for a few minutes. Adjust heat block to 90°C for DNA isolation. Add appropriate volume RLT and 100% EtOH to samples and mix thoroughly by inverting (RLT is equivolume to nuclei, but minimum is 320 uL; EtOH is 1.4x of nuclei volume, but minimum is 720 uL).
10. Transfer 700 uL of sample to a Qiagen RNeasy MinElute spin column. Spin 8000 g, 15 s, and dump flow-through.
11. Repeat step 42 until all of the sample has passed through column.
12. Add 350 uL Buffer FRN to column. Spin 8000 g, 15 s, and dump flow-through.
13. In a separate tube, combine 10 uL of DNaseI stock solution with 70 uL Buffer RDD per sample. Mix gently then briefly spin down.
14. Add the DNase incubation mix directly to the middle of the column and incubate at RT for 15 min.
15. Add 500 uL Buffer FRN to the column, spin 8000 g, 15 s. DO NOT DISCARD flow-through!
16. Place the column into a new 2 mL collection tube. Mix the flow-through by pipetting up and down a few times before transferring back to the column. Spin 8000 g, 15 s, and discard flow-through.

17. Add 500 uL Buffer RPE to the column, spin 8000 g, 15 s. Discard flow-through.
18. Repeat Step 49.
19. Spin column at max speed, 2 min to remove excess EtOH.
20. Transfer column to clean spin tube and add 14–30 uL RNase-free water directly to center of spin column. Incubate at RT, 2 min. Spin max speed, 1 min to elute RNA. Keep samples on ice after elution.
21. Quality control of RNA
    - Quantify RNA with Qubit. Qubit for RNA is not very sensitive, so will probably result in sample too low for most samples with not much nuclei, but useful to know concentration of other samples (e.g. unsorted nuclei will often generate a lot of RNA).
    - Run on Bioanalyzer Pico chip. The RIN of the samples will be terrible since the heat treatment shears RNA, but you are looking for a smear with a good representation of high molecular weight species rather than a band at the bottom indicating highly degraded RNA.
    - To minimize freeze-thaws, set aside RNA for cDNA production in PCR strip tubes. Ideally, start with 10 ng total RNA, but Ovation kit below accepts input from 500 pg - 100 ng.

VI. cDNA production using Nugen Ovation RNAseq V2 kit
    1. This is the exact protocol that Nugen provides. Nugen has a nice pdf that allows you to input the number of samples, and then automatically calculates correct amounts of solution for the three master mixes. Protocol is provided below for reference, but I prefer to just print out the adjusted pdf each time.
    2. All incubation steps are carried out using a thermocycler. For convenience, add an infinite incubation step set at the first incubation temperature before the actual program so that you can pre-heat/cool the thermocycler, add samples, then skip the infinite step.
    3. Before starting protocol, take RNAClean XP beads out of 4°C storage and let come to RT. Prepare fresh 70% EtOH. Once started, all steps should be completed on the same day. SPIA amplification (last) step can run O/N.
    4. Thaw RNA samples on ice. Add 2 uL of Primer mix (A1) to RNA. Gently mix by flicking, then briefly spin down. Incubate on thermocycler (65°C – 5 min, hold at 4°C).
    5. Remove tubes and place on ice. Per sample, add 3 uL of Master Mix A (2.5 uL Buffer Mix A2 +0.5 uL Enzyme Mix A3). Mix by flicking. Briefly spin down. Incubate on thermocycler (4°C – 1 min, 25°C – 1 min, 42°C – 10 min, 70°C – 15 min, hold at 4°C).
    6. Remove tubes and place on ice. Per sample, add 10 uL of Master Mix B (9.7 uL Buffer Mix B1 +0.3 uL Enzyme Mix B2). Mix by flicking. Briefly spin down. Incubate in thermocycler (4°C – 1 min, 25°C – 10 min, 50°C – 30 min, 80°C – 20 min, hold at 4°C).
    7. Mix RNAClean XP beads by inverting several times. Add 32 uL beads to each sample. Mix by pipetting 10 times. Incubate at RT 10 min.
    8. Place sample on magnetic stand and allow to clump, 5 min.
    9. Remove 45 uL of binding buffer.
    10. Add 200 uL of 70% EtOH and let stand for 30 s. Remove EtOH without disturbing beads.
    11. Repeat EtOH wash two more times.
    12. Remove all excess EtOH and let beads air dry on magnet for 5–15 min. All EtOH needs to be dry, but do not overdry beads. Look for beads that have just started cracking.
    13. Take tubes off magnet and add 40 uL of Master Mix C (20 uL Buffer Mix C2 +10 uL Primer Mix C1 +10 uL Enzyme Mix C3) to each tube. Pipette up and down to ensure the beads are well resuspended. Incubate in thermocycler (4°C – 1 min, 47°C – 60 min, 80°C – 20 min, hold at 4°C).
    14. Place tubes on magnetic stand and let clump 5 min.
    15. Remove sample and place in clean tube. Clean with Qiagen MinElute Reaction Cleanup Kit.
    16. Add to each sample 300 uL Buffer ERC. Mix by vortexing. Briefly spin down.
    17. Transfer to MinElute column. Spin 8000 g, 30 s. Discard flow-through.
    18. Wash with 700 uL Buffer PE. Spin 8000 g, 30 s. Discard flow-through.
    19. Dry column by spinning max speed, 1 min.
    20. Transfer column to new tube. Add 14–30 uL of Buffer EB directly to center of column. Let stand 1 min. Elute by spinning at max speed, 1 min.
    21. Assess cDNA quality using on TapeStation using a D1000 ScreenTape (3 uL of Sample Buffer + 1 uL sample/ladder). Like the RNA, cDNA should show up as a smear with lots of high molecular weight species.

22. Prepare 1:40 dilution of cDNA for qPCR. Check for enrichment/depletion of expected markers.
23. Quantify cDNA concentration using nanodrop.
24. Resuspend 100 ng-1ug of cDNA in 50 uL total volume of Illumina resuspension buffer (or TE ph8).

VII. Library construction for Illumina

1. Fill water tank (MilliQ water only) and turn on Covaris machine, let cool and degas for at least 40 min before use.
2. Transfer sample to Covaris MicroTUBE AFA Snap-Cap and place into machine.
3. Sonicate for 200 bp average size: Intensity 5, Duty Cycle 10%, Cycles per Burst 200, Treatment time 120 s.
4. Spin down briefly, then transfer to clean PCR strip tubes. Can store samples at −20°C.
5. For quality control, run a few samples on D1000 ScreenTape to verify fragmentation.
6. Start NEBNext library prep. Add to 50 uL fragmented DNA, 10 uL End Prep mix (7 uL Buffer Mix + 3 uL Enzyme Mix). Cap and invert tubes a few times to mix well. Spin briefly. Place on thermocycler with heated lid set to >75°C and incubate 20°C – 30 min, 65°C – 30 min, hold at 4°C.
7. Proceed directly to adapter ligation. Add to samples 31 uL of Ligation Mix (30 uL Ligation Master Mix + 1 uL Ligation Enhancer). Add 2.5 uL NEBNext Adapter for Illumina. Mix well by inverting tubes a few times. Spin briefly. Incubate at 20°C for 15 min in thermocycler with heated lid off.
8. Add 3 uL of USER Enzyme to samples. Mix by inverting and spin briefly. Incubate at 37°C for 15 min in thermocycler with heated lid set to >47°C. Samples can be stored at −20°C at this step.
9. Perform size selection with AMPure XP beads. Make sure to remove beads from 4°C storage and equilibrate to RT at least 30 min before starting. Make fresh 80% EtOH (same day). Perform two-sided selection as per NEB protocol (follow protocol for 200 bp insert). Resuspend beads by inverting several times right before use.
10. Add 40 uL (0.4x) XP beads and mix by pipetting up and down 10 times. Incubate samples at RT 5 min.
11. Place on magnetic stand and allow to clump for 5 min.
12. Carefully transfer supernatant to new tube. Discard beads with unwanted large fragments.
13. Add an additional 20 uL (0.2x) XP beads to transferred supernatant and mix by pipetting 10 times. Incubate samples at RT 5 min.
14. Place on magnetic stand and allow to clump for 5 min.
15. Remove and discard supernatant (small fragments) taking care not to disturb the beads.
16. Add 200 uL of 80% EtOH and let stand 30 s. Discard supernatant.
17. Repeat.
18. Make sure to remove all EtOH and let air dry 5–15 min. Do not overdry at this step. Elute when beads are still glossy looking.
19. Remove tubes from magnetic stand and elute in 15.5 uL of Illumina Resuspension Buffer. Mix well by pipetting.
20. Incubate at RT for 2 min then place back on magnetic stand for 2 min to clump.
21. Transfer 15 uL eluted DNA to new tube taking care not to transfer any beads.
22. Add to eluted DNA 35 uL of PCR mix (25 uL 2x Q5 Master Mix + 5 uL Universal Primer +5 uL Index Primer). Mix by inverting, then briefly spin. PCR conditions: 98°C – 30 s, (98°C – 10 s, 65°C – 75 s) x4-8 cycles, 65°C – 5 min, hold at 4°C.
23. Cleanup with XP beads. Add 45 uL (0.9x) XP beads and mix by pipetting.
24. Incubate RT 5 min, then place on magnetic stand and allow to clump for 5 min.
25. Add 200 uL of 80% EtOH and let stand 30 s. Discard supernatant.
26. Repeat.
27. Remove residual EtOH and air dry 5–15 min.
28. Add 31 uL of Illumina Resuspension Buffer and resuspend beads by pipetting. Incubate at RT for 2 min, then place on magnetic stand and let clump 2 min.
29. Remove 30 uL and transfer to clean tube.
30. Dilute library 1:5 then check size and concentration on TapeStation using High Sensitivity D1000 ScreenTape. Pool libraries.
31. Run another High Sensitivity D1000 ScreenTape with pooled samples to get average peak size and to check that no major pipetting errors occurred during pooling (concentration of pool should be around 10 nM).

32. Submit for sequencing with TapeStation results.

Chemicals from Sigma

- Sucrose, Tricine, spermidine trihydrochloride, spermine tetrahydrochloride, DTT, Tris, NaCl, KCl, $MgCl_2$, Glycerol, TritonX-100, NP-40
- Make stock solution to concentration indicated.

Other reagents

- Optiprep – Sigma D1556-250ML
- cOmplete mini protease inhibitors (EDTA free) – Roche distributed by Sigma 11836170001
- Superasin – Thermo AM2696
- RNasin – Promega N2515
- 16%formaldehyde solution – EMS 15710
- BSA – Jackson 001-000-162
- BSA RNase free – Thermo AM2618
- DyeCycle Ruby – Thermo V10273

Kits

- Qiagen AllPrep FFPE – Cat #80234
- Qiagen MinElute Reaction Cleanup – Cat #28206
- Nugen Ovation RNAseq
- NEBNext Ultra II DNA Library Prep Kit for Illumina - E7645L
- NEBNext Multiplex Adapters for Illumina – E7335L (Set 1) and E7500L (Set 2)
- Qubit dsDNA High Sensitivity - Q32854
- Qubit RNA HS - Q32855
- Bioanalyzer Pico chips - 5067–1513
- TapeStation D1000 - 5067–5582, ScreenTape; 5067–5583
- TapeStation High Sensitivity D1000 – 5067–5584, ScreenTape; 5067–5585

Equipment

- Dounce homogenizer – 40 mL all glass homogenizer with large and small clearance pestles – Kimble Chase 885300–0040
- Ultracentrifuge – Beckman Optima XE, Beckman SW41 Ti swinging rotor and buckets, Beckman Ultraclear 13.2 mL tubes (41121703)
- Refrigerated benchtop centrifuge similar to Eppendorf 5424R
- Nutrating Mixer
- FACS – BD FACSAria with 488, 561, 635/640 lasers
- Flow cytometry analyzer – BD LSR II with 488, 561, 640 lasers
- Software – Diva and FlowJo
- Agilent Bioanalyzer – with pico and nano chips for RNA
- Agilent TapeStation – D1000 and High Sensitivity D1000 screentapes and reagents
- Qubit Florometer
- Roche LightCycler 480
- Covaris S220 Focused Ultrasonicator
- Dynamags from Thermo – 96 side magnet 12331D

## Immunofluorescence

Mice were deeply anesthetized and transcardially perfused with phosphate buffered saline (PBS) followed by 4% formaldehyde (w/v) in PBS. Brains were dissected and postfixed overnight at 4°C, cryoprotected in 30% sucrose in PBS, OCT TissueTeck embedded, and cut with a Leica CM3050 S cryostat into 20 um sections that were directly mounted on slides. Slides were stored at −20°C. Antigen retrieval was performed by immersing slides into sodium citrate buffer (10 mM sodium citrate, 0.05% Tween 20, pH 6.0) at 95–100°C and simmering for 10 min in the microwave. The slides were cooled to room temperature and then washed with PBS. Immunofluorescence was performed by blocking for 30 min in IF Block Buffer (3% BSA in PBS with 0.1% TritonX-100), incubated with primary antibody overnight, washed three times with PBS, incubated with secondary antibody for one hour, washed with PBS, stained with DAPI solution (1 ug/mL in PBS) for 15 min, washed two times with PBS, and coverslipped with Prolong Diamond mounting media. For some samples, Tyramide signal

amplification was performed as follows: slides were incubated with secondary antibody conjugated to horseradish peroxidase for 1 hr, washed three times with PBS, incubated with Cy3 in Amplification Buffer from the TSA Cyanine three detection kit for 10 min, washed with PBS, and DAPI stained as above. All steps were performed at room temperature. Slides were imaged on a Zeiss LSM700 confocal microscope using the same acquisition settings for mouse and human slides. Brightness and contrast adjustments were made in ImageJ post-acquistion with the same adjustments applied to mouse and human images. In addition to the antibodies listed above for nuclei staining, the following primary antibodies were used for immunostaining:

## Primary antibodies

| Antigen | Species | Vendor | Cat. # | RRID | Dilution |
|---------|---------|--------|--------|------|----------|
| GFAP | Goat | Abcam | ab53554 | RRID:AB_880202 | 1:500 |
| Mog | Goat | Thermo | PA5-47319 | RRID:AB_2607363 | 1:500 |
| Mag | Rabbit | Cell Signaling | 9043S | RRID:AB_2665480 | 1:500 |
| Pde1a | Goat | Acris | TA311317 | | 1:200 |
| Pde1c | Rabbit | Santa Cruz | sc-376474 | RRID:AB_11149544 | 1:100 |

## Software used for analysis

Data processing steps made use of Linux tools in addition to custom perl scripts. In addition to standard Linux commands (e.g. awk, cut, sort, uniq), we used the following packages: trim_galore (v0.4.1) for adapter trimming of ATAC-seq samples, STAR (v2.4.2a) and Bowtie2 (v2.1.0) for read alignment, FastQC(0.11.4) and Picard (v1.123) for quality control metrics, SAMtools (v0.1.19–44428 cd) for indexing and removal of duplicates, igvtools (v2.3.32) for generating files for browser visualization, deepTools (v2.0) for analysis of ATAC-seq samples, MACS2 (v2.1.0) for peak calling of ATAC-seq samples, and featureCounts (v1.5.2) for read summerization. Data analysis was performed using R Studio (v 1.1.453, R: v3.5.0). In addition to base R (for data wrangling, Pearson correlations, hierarchical clustering, etc.), common R libraries, and custom R functions, we made extensive use of the following packages: tidyverse (v1.2.1.9000, which contains ggplot2, dplyr, etc.) for data analysis and visualization, DESeq2 (v1.21.6) for raw count normalization, differential expression analysis, and principal components analysis, Seurat (v2.3.3) for single nuclei/cell data filtering, normalization, and plotting, gplots (v3.0.1) and pheatmap (v1.0.10) for heatmaps, RColorBrewer (v1.1–2) for color palettes, GenomicScores (v1.2.2) for extracting conservation scores, and clusterProfiler (v3.9.1) for GO analysis. Command line parameters, custom R functions, and all R code used for analysis and for generating figures in this manuscript can be found at: https://github.com/xu-xiao/non_transgenic_cell_type_profiling; copy archived at https://github.com/elifesciences-publications/non_transgenic_cell_type_profiling (*Xu, 2018*).

## ATAC-seq read mapping, visualization, and analysis

Reads were processed with trim_galore with parameters '–stringency 3 –fastqc –paired'. Trimmed reads were mapped to mm10/hg38/rn6 using bowtie2 (*Langmead and Salzberg, 2012*) with parameters '-X 2000 –no-mixed –no-discordant'. Duplicates were removed using samtools (*Li et al., 2009*). Reads were normalized to 1x depth (reads per genome coverage, RPGC) and input using deepTools (*Ramírez et al., 2016*) bamCompare module, ignoring chrX, chrY, chrM and filtering reads for minimum mapping quality of 30. Metagene and heatmap profiles were generated using deepTools modules computeMatrix, plotProfile and plotHeatmap.

## Peak calling and peak analysis

Unique fragments under 100 nt were selected as they could not contain a nucleosome. These fragments were used to call peaks with MACS2 with parameters '–nomodel -q 0.01 –call-summits –keep-dup all'. The summits for each peak from the replicates were extended to 500 bp using bedtools slop. The peaks were filtered to remove chrY and chrM peaks and peaks that overlap with the mm10 or hg38 blacklist from ENCODE (*ENCODE Project Consortium, 2012*). Peaks overlapping with

genomic regions of interest were selected using bedtools intersect. Differentially accessible regions were identified using DiffBind and filtered for peaks with greater than two-fold change. Motifs were identified using findMotifGenome.pl from Homer with default parameters. SNP data was downloaded from the NHGRI-EBI GWAS catalog (v1.0.2, e93, r2018-08-14).

## Conservation analysis

To measure sequence conservation of ATAC-seq defined peaks, we used GenomicScores to extract PhastCons scores for each DNA base within a peak. 100-way PhastCons scores were used for human samples and 60-way PhastCons scores Conservation scores for mouse samples. For box plots, mean scores were computed for each promoter or gene body peak. Metagene profiles of conservation scores over promoters were generated with deepTools. Promoter peaks were defined as MACS2 identified peaks that overlap anywhere within 1 kb upstream to 100 bp downstream of the TSS. Gene body peaks were defined as MACS2 identified peaks located within the gene body of a gene, excluding any promoter peaks.

## RNA-seq quality control metrics, read mapping, visualization, and read summerization

Quality control of RNA-seq samples was performed using FastQC, and GC content was extracted for plotting. Reads were aligned using STAR (*Dobin et al., 2013*) and genome assemblies from ENSEMBL. In addition to default STAR parameters, we used the following parameters for single end (–outFilterMismatchNmax 999 –outFilterScoreMinOverLread 0 –outFilterMatchNminOverLread 0 – outFilterMatchNmin 35 –outFilterMismatchNoverLmax 0.05) and paired-end data (–outFilterMismatchNmax 999 –alignMatesGapMax 1000000 –outFilterScoreMinOverLread 0 –outFilterMatchNminOverLread 0 –outFilterMatchNmin 60 –outFilterMismatchNoverLmax 0.05). Aligned bam files were assessed for distribution of aligned reads using Picard CollectRnaSeqMetrics and then converted to tdf format for visualization using igvtools. Raw counts were generated using featureCounts (*Liao et al., 2014*). Refseq or ENSEMBL gene model annotations for whole genes were downloaded using the UCSC Table Browser tool. While ENSEMBL annotates more genes than Refseq (*Zhao and Zhang, 2015*), we found that visually, our mouse and human RNA-seq data better matches Refseq than ENSEMBL gene models. Because of this, for within species comparisons for mouse and human, we used Refseq annotations. Rat gene models are incompletely annotated by both Refseq and ENSEMBL, but as fewer genes are missing in the ENSEMBL annotation, we chose this annotation for within species comparisons. For comparative analysis across species, we used the ENSEMBL annotation for all species in order to match the list of orthologous transcripts that we obtained from ENSEMBL. The following parameters were used in addition in addition to the default in feature-Counts: for paired-end data, fragments are counted instead of reads (-p), chimeric fragments are not counted (-C). For comparative analysis across species, reads are allowed to be assigned to more than one matched meta-feature (-O) in order to avoid problems when genes overlap in one species but not in another. For within-species comparisons, reads that map to more than one matched meta-feature are not counted (default).

## Genome/gene annotations

| Purpose | Program | Species | Annotations | Source | Genome |
|---|---|---|---|---|---|
| Genome alignment | STAR/Bowtie2 | Mouse | ENSEMBL | ENSEMBL | mm10 |
| Genome alignment | STAR/Bowtie2 | Rat | ENSEMBL | ENSEMBL | rn6 |
| Genome alignment | STAR/Bowtie2 | Human | ENSEMBL | ENSEMBL | hg38 |
| Read quantification | featureCounts | Mouse | Refseq | UCSC | mm10 |
| Read quantification | featureCounts | Rat | ENSEMBL | UCSC | rn6 |
| Read quantification | featureCounts | Human | Refseq | UCSC | hg38 |
| Ortholog read quantification | featureCounts | Mouse | ENSEMBL | ENSEMBL | mm10 |
| Ortholog read quantification | featureCounts | Rat | ENSEMBL | ENSEMBL | rn6 |
| Ortholog read quantification | featureCounts | Human | ENSEMBL | ENSEMBL | hg38 |

## Generation of ortholog annotations

Because we were interested in differences in highly conserved cell types, we only wanted to study differences in high confidence orthologs. Schematics for generating annotations for mouse-rat-human orthologs, and mouse-human orthologs are shown in *Figure 3—figure supplement 1A* and *2A* respectively. In both cases, we downloaded a list of orthologous transcripts from the ENSEMBL BioMart (release 88). First, we removed any annotation that contained a low confidence pair (ENSEMBL orthology confidence = 0). Next, we removed duplicate annotations as well as any orthologs that were not strictly 1:1, which we defined as transcript that is annotated to be orthologous to more than one transcript in another species. To avoid any differential expression simply due to changes in gene length, we filtered out any genes that change in length by more than two-fold across any species. Additionally, as we were not interested in examining small RNAs, we also filtered out genes smaller than 1 kb in genome length (including untranslated regions). Finally, because we were interested in gene level rather than transcript level differences, we used the annotation for the longest transcript as the gene annotation. Because the rat genome is incompletely annotated, the list of orthologous genes between mouse and human contains around 3,000 more genes than the list of orthologous genes between mouse, rat, and human.

## Gene count normalization, differential expression analysis, and principal components analysis

Raw gene count tables were loaded into R and normalization and differential expression analysis was performed using DESeq2 (*Love et al., 2014*). For age and PMD, numerical covariates were used for modeling. Statistically significant genes were defined as having an adjusted p-value of less than 0.01. Fold change cutoffs were not used unless otherwise noted. The rlog function was used to generate normalized counts for downstream analysis (e.g. heatmaps, clustering). A modified version of the plotPCA function was used to perform principal components analysis and to return both scores and loadings. Analysis of samples from mouse and human revealed no significant batch effects (*Figure 3E*). Additionally, no significant differences in GC content between the RNAseq libraries (*Supplementary file 1*, *Figure 2—figure supplement 1D*), or relationships between changes in gene expression and gene length or GC content were observed (*Figure 3—figure supplement 2*).

## Hierarchical clustering

Hierarchical clustering was performed using normalized gene counts produced by the rlog function from DESeq2 (*Love et al., 2014*). Unless specified, the top 250 most variable genes across samples, as computed by the rowVars function, were used for clustering. We use the most variable genes instead of all genes for clustering because in our experience, for comparisons of related cell types such as neuroglia, the most variable genes are the most informative for defining the different functions between cell types. Additionally, our finding that gene expression between different cell types is highly correlated (r = 0.91–0.94 for mouse cerebellar cell types, *Figure 1G*) suggests that clustering using all genes might dilute the strong but relatively few gene expression differences between cell types. This is especially true for comparisons across species - although we tried to compile an annotation consisting of only high confidence orthologous genes, it is possible that small differences in annotation across species can accumulate especially over genes that are expressed at low levels in both species. Default parameters (Euclidean distance and complete linkage) were used with the dist and hclust functions to produce clusters.

## Specificity index algorithm

The specificity index (SI) algorithm was previously developed in the lab to find genes that are specific in a dataset of interest as compared to other datasets (*Dougherty et al., 2010*). Because the previous algorithm was designed for microarray data, we adapted it to accept RNA-seq data as input. We have also added a feature that incorporates information from biological replicates for calculations. In our implementation, FPKMs are calculated from raw counts to avoid biases toward long genes, and genes less than 1 kb are excluded to avoid biases from extremely short genes. Next, FPKM values are logged and $\log_{10}$(FPKM) values are bottomed out at 0 to avoid biases from long,

non-expressed genes. To take into account sample replicates, we calculate the mean and standard deviation for all genes within a cell type. SI values are calculated as described previously, but instead of using mean gene expression values as input, we randomly sample from a normal distribution with the mean and standard deviation of gene expression. This process is performed 1000 times and final ranks are averaged across all iterations. R code containing this algorithm can be found at: https:// github.com/xu-xiao/non_transgenic_cell_type_profiling; copy archived at https://github.com/elifes-ciences-publications/non_transgenic_cell_type_profiling (*Xu, 2018*). For analysis of the distribution of highly specific genes in the mouse cerebellum, images for each gene from the Allen Mouse Brain Atlas (*Lein et al., 2007*) were examined. Genes that were not in the Atlas at the time of analysis, or genes that showed no staining in any region of the brain were excluded from analysis. The percentage of genes showing proper distribution for the cell type analyzed were calculated.

## GO analysis

Significant differentially expressed genes were split into up- and down-regulated (or mouse and human enriched) genes and gene ontology enrichment analysis for each group was performed using clusterProfiler (*Yu et al., 2012*). Significant ontologies were defined as having a q-value of less than 0.01 (default). *Supplementary file 5* contains all GO analysis results, including significant ontologies from any of the three domains (biological process, molecular function, cellular component). For simplicity of visualization, only biological process ontologies were used. Network based CNET plots were generated using the cnetplot function.

## Comparison with single nuclei RNA-seq (sNuc-Seq) data

For analysis of sequencing depth and number of genes detected, we compared our data to mouse single nuclei RNA-seq (sNuc-Seq) data from (*Habib et al., 2016*), which was generated using the Smart-seq2 platform. Because Smart-seq2 generates libraries from full length cDNA, we were able to directly compare these datasets to our bulk nuclear profiles for the same cell types, by download-ing raw sequence files (GSE85721), aligning to the mm10 genome, and generating raw counts against mouse Refseq whole gene annotation as described above. We obtained sNuc-Seq data for 17 astrocytes, eight oligodendrocytes, and 7 OPCs. Log transformed transcripts per million (TPMs) were calculated for both sNuc-Seq and our nuclear profiles as described by (*Habib et al., 2016*) except that lengths for the whole gene instead of transcripts (exons only) were used for normaliza-tion. To explore the effect of sequencing depth on the number of genes detected, the number of mapped reads for single nuclei data were extracted using samtools flagstat. We next used samtools view to randomly downsample our nuclear data to 50, 100, 200, 300, 400, 500, 600, 800, 1,000, and 1200 thousand reads. We regenerated gene counts using the downsampled data and calculated TPMs. Using the same threshold as (*Habib et al., 2016*), we considered a gene to be expressed if it has a $\log_2(\text{TPM} + 1) > 1.1$.

For expression analysis of species-enriched genes, we downloaded mouse cerebellar single nuclei RNA-seq data from (*Lake et al., 2018*) and human cerebellar single cell RNA-seq data from (*Saunders et al., 2018*). Because the libraries generated in these studies are different from our libraries (e.g. Lake et al. used PolyA selection on their nuclear RNA and Saunders et al. uses RNA from whole cells), we used author provided count matrices (files described in *Supplementary file 1*) rather than reprocessing their data. We also used author provided cluster annotations. We used Seurat in R to produce violin plots and to obtain normalized counts. Genes detected in fewer than three cells and cells with fewer than 300 or more than 5000 detected genes were omitted. We used global scaling to normalize the gene expression measurements for each cell by the total expression, multiplied by a scale factor of 10,000, and log-transform the result. Normalized expression was used for violin plots. For ECDF plots, for each SN-seq defined cell type, we extracted normalized expres-sion for human-enriched, mouse-enriched, or housekeeping genes (genes obtained from species comparative analysis, as described above) in all samples corresponding to that cell type. For each gene, mean expression across samples or proportion of samples with expression above 1 $\log_2(\text{TPM})$ were calculated plotted.

## Acknowledgments

We thank members of the Heintz lab for helpful discussion, suggestions, and improvements. We thank Eric Schmidt, Dakota Blackman and Winrich Freiwald for the rat tissue. We thank Svetlana Mazel, Selamawit Tadesse, Stanka Semova, Songyan Han, and Xiao Li from The Rockefeller University Flow Cytometry Resource Center. We thank Connie Zhao and The Rockefeller University Genomics Resource Center.

## Additional information

### Competing interests

Xiao Xu: XX is named as an inventor on a patent application filed on this technology, assigned to The Rockefeller University (application number PCT/US2018/017556). XX is currently an employee at a startup company using this technology. Conduct of the studies presented in this manuscript were not influenced by these relationships. Nathaniel Heintz: NH is named as an inventor on a patent application filed on this technology, assigned to The Rockefeller University (application number PCT/US2018/017556). NH is a founder of a startup company using this technology. Conduct of the studies presented in this manuscript were not influenced by these relationships. The other authors declare that no competing interests exist.

### Funding

| Funder | Grant reference number | Author |
| --- | --- | --- |
| Howard Hughes Medical Institute | | Nathaniel Heintz |
| National Institutes of Health | 1P30 DA035756-01 | Nathaniel Heintz |
| Leon Black Family Foundation | | Nathaniel Heintz |
| Rockefeller University | Robertson Therapeutic Development Fund | Xiao Xu Nathaniel Heintz |
| Estelle G. Kestenbaum Award for Innovative Research in Neurodegenerative Disease | | Xiao Xu |

The funders had no role in study design, data collection and interpretation, or the decision to submit the work for publication.

### Author contributions

Xiao Xu, Conceptualization, Data curation, Software, Formal analysis, Supervision, Funding acquisition, Validation, Investigation, Visualization, Methodology, Writing—original draft, Writing—review and editing; Elitsa I Stoyanova, Formal analysis, Validation, Investigation, Visualization, Writing—review and editing; Agata E Lemiesz, Investigation, Writing—review and editing; Jie Xing, Validation, Investigation; Deborah C Mash, Resources, Writing—review and editing; Nathaniel Heintz, Conceptualization, Resources, Supervision, Funding acquisition, Methodology, Writing—original draft, Project administration, Writing—review and editing

### Author ORCIDs

Xiao Xu https://orcid.org/0000-0001-9612-5569
Elitsa I Stoyanova http://orcid.org/0000-0001-6400-6119
Agata E Lemiesz http://orcid.org/0000-0002-8237-975X
Nathaniel Heintz https://orcid.org/0000-0002-8874-8704

### Ethics

Animal experimentation: All animal protocols were carried out in accordance with the US National Institutes of Health Guide for the Care and Use of Laboratory Animals and were approved by The

Rockefeller University Institutional Animal Care and Use Committee. Animal protocols (14680-H and 17011) were approved by The Rockefeller University Institutional Animal Care and Use Committee.

## Decision letter and Author response
Decision letter https://doi.org/10.7554/eLife.37551.040
Author response https://doi.org/10.7554/eLife.37551.041

## Additional files

### Supplementary files
• Supplementary file 1. Summary of all RNA-seq datasets, including information about animals, sorts, and quality control metrics related to methods. Also includes information on published RNA-seq datasets used in this manuscript.
DOI: https://doi.org/10.7554/eLife.37551.019

• Supplementary file 2. Clinical information for all human tissue donors, related to methods.
DOI: https://doi.org/10.7554/eLife.37551.020

• Supplementary file 3. Specificity index calculations for mouse, rat, and human cell types using either species-specific annotations or with mouse-rat-human orthologous gene annotations. SIs for mouse and rat samples with orthologous gene annotations have been calculated using either all cell types (all) or without Purkinje samples.
DOI: https://doi.org/10.7554/eLife.37551.021

• Supplementary file 4. Differential expression analysis results for mouse and human-enriched genes. Also mouse and human IDs for genes that are unchanged in gene expression between the two species.
DOI: https://doi.org/10.7554/eLife.37551.022

• Supplementary file 5. Annotation of ATAC peaks. All: all MACS2 called peaks; DA: differentially accessible peaks between granule and basket neurons.
DOI: https://doi.org/10.7554/eLife.37551.023

• Supplementary file 6. Motif analysis of various ATAC-seq defined regions. DA: regions that are differentially accessible between granule (gran) and basket (bsk) neurons in mouse (m) or human (h); HE: regions defined by peaks located in the promoter (p) or gene body (gb) of human (h) enriched genes for granule (gran) or basket (bsk) neurons; ME: regions defined by peaks located in the promoter (p) or gene body of mouse (m) enriched genes for granule (gran) or basket (bsk) neurons.
DOI: https://doi.org/10.7554/eLife.37551.024

• Supplementary file 7. Differentially accessible regions between human granule and basket neurons that contain single nucleotide polymorphisms (SNPs) associated with human disease. The column Multiple specifies whether a SNP has been linked to a specific disease/trait in at least two publications.
DOI: https://doi.org/10.7554/eLife.37551.025

• Supplementary file 8. Differential expression analysis results for the influence of clinical factors on gene expression in human samples.
DOI: https://doi.org/10.7554/eLife.37551.026

• Supplementary file 9. Full results from all gene ontology (GO) analyses performed in the paper.
DOI: https://doi.org/10.7554/eLife.37551.027

• Transparent reporting form
DOI: https://doi.org/10.7554/eLife.37551.028

### Data availability
A summary of all sequencing data can be found in Table S1. All sequencing data have been deposited in GEO under accession code GSE101918.

The following dataset was generated:

Database, license,
and accessibility

| Author(s) | Year | Dataset title | Dataset URL | information |
|---|---|---|---|---|
| Xiao Xu, Elitsa I Stoyanova, Agata E Lemiesz, Jie Xing, Deborah C Mash, Nathaniel Heintz | 2017 | Species and Cell-Type Properties of Classically Defined Human and Rodent Neurons and Glia | https://www.ncbi.nlm. nih.gov/geo/query/acc. cgi?acc=GSE101918 | Publicly available at the NCBI Gene Expression Omnibus (accession no. GSE10 1918) |

The following previously published datasets were used:

| Author(s) | Year | Dataset title | Dataset URL | Database, license, and accessibility information |
|---|---|---|---|---|
| Mo A, Mukamel EA, Davis FP, Luo C, Eddy SR, Ecker JR, Nathans J | 2015 | Epigenomic Signatures of Neuronal Diversity in the Mammalian Brain | https://www.ncbi.nlm. nih.gov/geo/query/acc. cgi?acc=GSE63137 | Publicly available at the NCBI Gene Expression Omnibus (accession no. GSE63137) |
| Habib N, Li Y, Heidenreich M, Swiech L, Avraham-Davidi I, Trombetta JJ, Hession C, Zhang F, Regev A | 2016 | Div-Seq: Single-nucleus RNA-Seq reveals dynamics of rare adult newborn neurons | https://www.ncbi.nlm. nih.gov/geo/query/acc. cgi?acc=GSE84371 | Publicly available at the NCBI Gene Expression Omnibus (accession no. GSE84371) |
| Lake BB, Chen S, Sos BC, Fan J, Yung YC, Chun J, Kharchenko PV, Zhang K | 2018 | Integrative single-cell analysis of transcriptional and epigenetic states in the human adult brain [snDrop-seq] | https://www.ncbi.nlm. nih.gov/geo/query/acc. cgi?acc=GSE97930 | Publicly available at the NCBI Gene Expression Omnibus (accession no. GSE97930) |
| Saunders A, McCarroll S | 2018 | A Single-Cell Atlas of Cell Types, States, and Other Transcriptional Patterns from Nine Regions of the Adult Mouse Brain | https://www.ncbi.nlm. nih.gov/geo/query/acc. cgi?acc=GSE116470 | Publicly available at the NCBI Gene Expression Omnibus (accession no. GSE116470) |

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
