## [Decision Letter]

Thank you for submitting your article "Species and Cell-Type Properties of Classically Defined Human and Rodent Neurons and Glia" for consideration by *eLife*. Your article has been reviewed by three peer reviewers, one of whom is a member of our Board of Reviewing Editors, and the evaluation has been overseen by a Senior Editor. The reviewers have opted to remain anonymous.

The reviewers have discussed the reviews with one another and the Reviewing Editor has drafted this decision to help you prepare a revised submission.

Summary:

In this manuscript, the authors expand the repertoire of FACS-based nuclear isolation for cell-type specific gene expression studies by using antibodies against plasma membrane proteins and ER proteins (in addition to nuclear proteins) to purify nuclei from populations of neurons and non-neuronal cells from rodent and human cerebellum. They compare their FANS data against a genetically-based nuclear sorting method and they make cross-species comparisons to validate the utility of the specific antibody combinations. Finally they use this FANS method to purify specific cell types from human cerebellum for RNAseq and ATACseq, and perform comparative studies to assess gene expression differences with respect to biological features (i.e. age, sex) of the samples. They argue that this method will enhance utility of FACS-based nuclear isolation of specific cell types from brain tissue of species (e.g. rat, human) for which genetic tagging is not feasible. They demonstrate this by comparing cerebellar cell type gene expression from 16 postmortem human brains.

The manuscript contains high quality data and we agreed with the authors that the idea of applying additional antibodies to FACS methods for species that are not amenable to genetic tagging (especially human) has the potential to be useful in ways that may lead to other discoveries in the future. However none of the findings of the manuscript or the approach itself are entirely novel. In addition the reviewers engaged in a long and productive discussion about the likely impact of this study as compared with single cell sequencing analyses in light of the heterogeneity that is likely to exist within each of the FANS purified populations. We all felt that some additional experimental evidence and substantial changes in the text of the manuscript would be needed to address these concerns, yet in the end we concluded that the manuscript provides a great platform for the discussion of the value of single cell versus population methods.

With these ideas in mind, we hope the authors will be able to revise the article in the areas suggested below to maximize its impact to the field.

Essential revisions:

*1) Redirect to the Tools and Resources section of the journal.*

Because the datasets are the central part of the manuscript more than the conclusions, we suggest the manuscript be directed to the "Tools and Resources" section of the journal, because this article fits well the description of offering "substantial improvements and extensions of existing technologies" as well as having the "potential to facilitate experiments that address problems that to date have been challenging or even intractable."

2) Address analytical consequences of heterogeneity in the populations isolated by FANS.

The reviewers discussed at length the possibility that heterogeneity within the purified populations could confound some of the outcomes of the analysis. For example, if the population of cells the authors call "astrocytes" based on their FANS profile in mice are actually comprised of more than one cell type and the population of "astrocytes" purified by FANS from human are also comprised of more than one cell type that overlaps but is not identical to the population from mouse, then these populations will be said to have species-specific gene expression that may instead simply reflect the differential FANS enrichment of portions of a heterogeneous cell population. The text of the manuscript needs to be adapted to more directly address this alternative explanation of the data throughout (not just for this experiment). In addition, some additional experimental data should be added to address these concerns. Ideally, the authors could directly compare bulk sequencing from FANS purified cells with scRNA-seq of the same purified population to get a measure of heterogeneity within the purified populations for experiments that provide key conclusions in the study. However we appreciate that this experiment may be beyond the scope of the current study, and agree that it is not necessary for revision of the manuscript (though it would be a great service to the community discussion if the authors were able to perform this comparison). Alternatively, the authors could use a follow up experimental validation of some of the conclusions of the comparative part of the study. This should go beyond the few obvious examples (totally cell-type or species-specific genes) because presumably the bioinformatic differences assessed at the statistical level are driven by differences in the levels of expression of sets of genes. We ask the authors to determine what data they feel can be presented to best address this concern, which was a major point of the reviewers' discussion during the review process.

3) Enhance information on validation of the FACS antibodies.

More details need to be provided regarding the specificity and reactivity of the antibodies chosen by the authors. For example, what quantitative or experimental guidance was used to select the antibody dilutions used for FANS? Given the authors' description of the variable detection of NeuN protein (but not RNA) in the mouse and rat cerebellar neural subtypes, this may be a consequence and could have significant implications for the purity, and relative homogeneity of their enriched populations. Furthermore, given the variable nature of cross-species reactivity of antibodies, differences in antibody affinities for subpopulations of cell types such as astrocytes could underlie the apparent differences in gene expression by species as suggested above.

4) Strengthen the depth of the ATAC-seq data.

Given that this is one of the more novel sections of the manuscript, the authors should deepen the ATAC-seq analyses. First, because the isolation of nuclei involves formaldehyde fixation (1% PfA for 8 min at room temperature), which is not widely used in ATAC-seq preparations, it is important to experimentally validate whether data quality of cell-type-specific ATAC-seq using fixed nuclei is comparable to standard ATAC-seq assays using unfixed nuclei. For instance, the authors can perform a pair-wise comparison between fixed and unfixed nuclei isolated from mouse cerebellum, so we can assess the data quality of the cell-type specific ATAC-seq data generated from fixed nuclei. Second, the authors can deepen the computational analysis of FANS-based ATAC-seq datasets. For instance, the authors should cross-reference all cell-type-specific ATAC-seq peaks with annotated genomic features (promoters, intergenic regions and gene bodies). Also, TF motifs enriched at these cell-type-specific open chromatin regions should be analyzed and reported. Lastly, it would be good if the authors could discuss the value of the putative cell-type-specific gene regulatory regions (ATAC-seq peaks) in human cerebella to investigators studying human diseases and identifying SNPs or DNA variants (within non-coding regions). Cross-referencing these two data sets might help pinpoint variants of functional significance.

5) Walk back conclusions of the stress section.

The final section on the induction of stress responses does not significantly contribute to the study as written and is not experimentally testable. One possible alternative explanation is that induction of IEGs could have arisen from tissue processing (as has been reported in other studies such as PMID 29230054) rather than prior to death. In fact, given the nice effort of this manuscript in discussing the technical details of the methods, this would likely be a better use of this section than speculation about the use of IEG profiles to propose possible pathological insults not evident from the medical information provided with these brains. Another possibility is that the data could arise from trying too hard to find statistical patterns in the data. The entire 'stress' analysis is predicated on the observation that three samples expressed high levels of a single gene. The authors use this justification to cluster these samples together and look for differential genes between this group and the remaining samples. Would the authors find a similar number of differentially expressed genes if they were to shuffle the sample labels and bootstrap this analysis using random sets of three samples?

---

## [Author Response]

Essential revisions:

1) Redirect to the Tools and Resources section of the journal.

Because the datasets are the central part of the manuscript more than the conclusions, we suggest the manuscript be directed to the "Tools and Resources" section of the journal, because this article fits well the description of offering "substantial improvements and extensions of existing technologies" as well as having the "potential to facilitate experiments that address problems that to date have been challenging or even intractable."

This is fine with us. We have directed the manuscript to the Tools and Resources” section.

2) Address analytical consequences of heterogeneity in the populations isolated by FANS.

The reviewers discussed at length the possibility that heterogeneity within the purified populations could confound some of the outcomes of the analysis. For example, if the population of cells the authors call "astrocytes" based on their FANS profile in mice are actually comprised of more than one cell type and the population of "astrocytes" purified by FANS from human are also comprised of more than one cell type that overlaps but is not identical to the population from mouse, then these populations will be said to have species-specific gene expression that may instead simply reflect the differential FANS enrichment of portions of a heterogeneous cell population. The text of the manuscript needs to be adapted to more directly address this alternative explanation of the data throughout (not just for this experiment). In addition, some additional experimental data should be added to address these concerns. Ideally, the authors could directly compare bulk sequencing from FANS purified cells with scRNA-seq of the same purified population to get a measure of heterogeneity within the purified populations for experiments that provide key conclusions in the study. However we appreciate that this experiment may be beyond the scope of the current study, and agree that it is not necessary for revision of the manuscript (though it would be a great service to the community discussion if the authors were able to perform this comparison). Alternatively, the authors could use a follow up experimental validation of some of the conclusions of the comparative part of the study. This should go beyond the few obvious examples (totally cell-type or species-specific genes) because presumably the bioinformatic differences assessed at the statistical level are driven by differences in the levels of expression of sets of genes. We ask the authors to determine what data they feel can be presented to best address this concern, which was a major point of the reviewers' discussion during the review process.

In response to the reviewers’ request that we directly compare our data to single nucleus data from the same cerebellar cell types, we have provided an extensive comparative analysis of our data with published snRNAseq data from mouse (Saunders et al., 2018) and human (Lake et al., 2018) cerebellum. This was possible because we focused in the original manuscript (and in this comparison) on very highly and differentially expressed orthologous genes, allowing direct comparison despite the reduced sensitivity of ssRNAseq data. These additional data are presented in Figures 3—figure supplement 2 and 3, and they allow several additional points to be made in the extensively revised section “Cross-species comparison reveals cell-type and species specific genes”. First, and most important, the species specific changes in expression we observed in our data are also evident in the re-analysis of the mouse and human ssRNAseq data. We provide direct measurements of expression of a variety of these genes from the published single nuclei data (Figure 3—figure supplement 3A). Furthermore, plots of mean expression level and proportion of expressing nuclei confirm quantitatively that the species specific expression of the entire class is also evident in the single nucleus data. Second, the differences between mouse and human expression cannot have arisen in our data as a consequence of the sorting of different cell populations because lack of expression in either human or mouse is observed for all categories of cell types that arose in clustering the single cell datasets (Figure 3—figure supplement 3C, D). This is also evident for specific examples in the plots presented in Figure 3—figure supplement 3A and the additional data presented in Figure 4—figure supplement 2. Third, we find that Lake et al. have misinterpreted their single nucleus data to include two populations of Purkinje cells. The expression profiles we have generated, and the expression of classical markers for these cell types, establish that these clusters are actually cerebellar interneurons rather than Purkinje cells. We do not note this error to criticize the published data. Rather, proper classification of the single nucleus data was necessary for us preform the direct comparisons that have been included in this revision.

3) Enhance information on validation of the FACS antibodies.

More details need to be provided regarding the specificity and reactivity of the antibodies chosen by the authors. For example, what quantitative or experimental guidance was used to select the antibody dilutions used for FANS? Given the authors' description of the variable detection of NeuN protein (but not RNA) in the mouse and rat cerebellar neural subtypes, this may be a consequence and could have significant implications for the purity, and relative homogeneity of their enriched populations. Furthermore, given the variable nature of cross-species reactivity of antibodies, differences in antibody affinities for subpopulations of cell types such as astrocytes could underlie the apparent differences in gene expression by species as suggested above.

We have added further description of the antibodies, staining, and gating strategies in three different areas:

- We have added a supplemental document containing a detailed protocol, which has guidelines on the amount of starting tissue, since the density of nuclei in the staining reaction is as important as the antibody concentration.

- In the first paragraph under the section "Antibody-based sorting of cell types is easily transferrable to other species", we added description of how we identify and validated distinct populations in rat.

- In Materials and methods under the section "Nuclei labeling and sorting", we added information on how we determined antibody concentrations, acknowledging that we do see differences in cross-species reactivity of antibodies. We have tried to make to clear that although gated populations do not look exactly the same by FACS in the three species, they are nonetheless distinct and reproducible across experiments within the same species.

In regards to purity, we rigorously validate that each sample contains the expected markers for the target cell type and does not contain markers for other cell types. We hope that the examples of this validation (e.g. Figures 1C, 1D, 1B, Figure 2—figure supplement 1B, Figure 5B), the high correlation between samples containing the same cell type (e.g. Figures 1G, Figure 2D, Figure 5C), and our ability to identify genes including known markers that are highly specific to each cell type using the specificity index (Figures 2E, Figure 2—figure supplement 1A, Supplementary file 3) are indicative of the high purity of our sorts.

Finally, in regards to the differences in species gene expression, we hope that our additional experiments as described in response to point 2 sufficiently addresses this issue.

4) Strengthen the depth of the ATAC-seq data.

Given that this is one of the more novel sections of the manuscript, the authors should deepen the ATAC-seq analyses. First, because the isolation of nuclei involves formaldehyde fixation (1% PfA for 8 min at room temperature), which is not widely used in ATAC-seq preparations, it is important to experimentally validate whether data quality of cell-type-specific ATAC-seq using fixed nuclei is comparable to standard ATAC-seq assays using unfixed nuclei. For instance, the authors can perform a pair-wise comparison between fixed and unfixed nuclei isolated from mouse cerebellum, so we can assess the data quality of the cell-type specific ATAC-seq data generated from fixed nuclei. Second, the authors can deepen the computational analysis of FANS-based ATAC-seq datasets. For instance, the authors should cross-reference all cell-type-specific ATAC-seq peaks with annotated genomic features (promoters, intergenic regions and gene bodies). Also, TF motifs enriched at these cell-type-specific open chromatin regions should be analyzed and reported. Lastly, it would be good if the authors could discuss the value of the putative cell-type-specific gene regulatory regions (ATAC-seq peaks) in human cerebella to investigators studying human diseases and identifying SNPs or DNA variants (within non-coding regions). Cross-referencing these two data sets might help pinpoint variants of functional significance.

Although formaldehyde fixed nuclei have not been widely used in ATAC-seq preparations, we are using a modified protocol that was developed specifically for fixed nuclei by the Greenleaf and Chang labs, the groups that initially developed ATAC-seq (Chen et al., 2016). The 2016 publication from these groups does fairly extensive validation showing that fixation (1% PFA for 10 minutes at room temperature) does not affect tagmentation efficiency and that data generated from fixed cells are comparable to standard conditions using living cells (R = 0.93). Our Materials and methods section already cites this paper and contains the version of protocol for fixed nuclei. We have updated it to make it clear that we are using the modified rather than standard protocol, have also added a sentence to the Results section to clarify this point.

To strengthen the depth of the ATAC section of the paper, we have performed several additional analyses, and added additional figures (Figure 4—figure supplement 1, Figure 4—figure supplement 2D-E) and tables (Supplementary files 5-7). These include:

A) Annotation of all ATAC peaks, including differentially accessible peaks between granule and basket neurons, with genomic features (Supplementary file 5, Figure 4—figure supplement 1A).

B) Validated that cell-type specific ATAC-seq derived regulatory regions is associated with cell-type specific gene expression. This was performed in two ways:

- From genes that were identified to by RNA-seq be differentially expressed between granule and basket neurons, analysis of ATAC coverage over promoters and gene bodies (Figure 4—figure supplement 1B, C).

- From regions that were identified by ATAC-seq to be differentially accessible between granule and basket neurons, analysis of expression of genes associated with different classes of peaks (Figure 4—figure supplement 1D).

- Motif analysis in each cell type of either all peaks or differentially accessible peaks (Supplementary file 6, Figure 4—figure supplement 2D). From this analysis, we identified GATA motifs in regions that are differentially accessible in mouse granule neurons, and a GATA transcription factor that is expressed in mouse but not human granule neurons.

- Cross-referencing of human ATAC peak regions with NHGRI-EBI GWAS SNP catalog (Supplementary file 7, Figure 4—figure supplement 2E). Mainly, this analysis can serve as a reference for investigators studying human diseases, although we have highlighted an interesting example of a SNP associated with Parkinson's.

5) Walk back conclusions of the stress section.

The final section on the induction of stress responses does not significantly contribute to the study as written and is not experimentally testable. One possible alternative explanation is that induction of IEGs could have arisen from tissue processing (as has been reported in other studies such as PMID 29230054) rather than prior to death. In fact, given the nice effort of this manuscript in discussing the technical details of the methods, this would likely be a better use of this section than speculation about the use of IEG profiles to propose possible pathological insults not evident from the medical information provided with these brains. Another possibility is that the data could arise from trying too hard to find statistical patterns in the data. The entire 'stress' analysis is predicated on the observation that three samples expressed high levels of a single gene. The authors use this justification to cluster these samples together and look for differential genes between this group and the remaining samples. Would the authors find a similar number of differentially expressed genes if they were to shuffle the sample labels and bootstrap this analysis using random sets of three samples?

We have rewritten this section to deemphasize the stress response and instead use it as an example of gene expression changes that are not explained by clinical data. We have added the possibility that these changes may arise due to agonal factors or tissue processing. Finally, we performed resampling to show that the number of genes that are differentially expressed in the three individuals are in the top 1% of the distribution of any random set of three samples (see Figure 7—figure supplement 1).